# GEOMETRIC SIGNATURES OF COMPOSITIONALITY ACROSS A LANGUAGE MODEL'S LIFETIME

## ABSTRACT

Compositionality, the notion that the meaning of an expression is constructed from the meaning of its parts and syntactic rules, permits the infinite productivity of human language. For the first time, artificial language models (LMs) are able to match human performance in a number of compositional generalization tasks. However, much remains to be understood about the representational mechanisms underlying these abilities. We take a high-level geometric approach to this problem by relating the degree of compositionality in a dataset to the intrinsic dimensionality of its representations under an LM, a measure of feature complexity. We find not only that the degree of dataset compositionality is reflected in representations' intrinsic dimensionality, but that the relationship between compositionality and geometric complexity arises due to learned linguistic features over training. Finally, our analyses reveal a striking contrast between linear and nonlinear dimensionality, showing that they respectively encode formal and semantic aspects of linguistic composition.

## 1 INTRODUCTION

By virtue of linguistic compositionality, few syntactic rules and a finite lexicon can generate an unbounded number of sentences (Chomsky, 1957). That is, language, though seemingly high-dimensional, can be explained using relatively few degrees of freedom. A great deal of effort has been made to test whether neural language models (LMs) exhibit human-like compositionality (Hupkes et al., 2019; Baroni, 2019; McCoy, 2022). We take a geometric view of this question, asking how an LM's representational structure reflects and supports compositional understanding over training.

If a language model is a good model of language, we expect its internal representations to reflect the relatively few variables underlying the latter. That is, representations should reflect the *manifold hypothesis*, or the notion that real-life, high-dimensional data lie on a low-dimensional manifold (Goodfellow et al., 2016). The dimension of this manifold, or *intrinsic dimension* (ID), is then the minimal number of degrees of freedom required to describe it without suffering from information loss (Goodfellow et al., 2016; Campadelli et al., 2015). The manifold hypothesis has indeed been attested for linguistic representations: LMs have been found to compress inputs to an ID orders-of-magnitude lower than their extrinsic dimension (Cai et al., 2021; Cheng et al., 2023; Valeriani et al., 2023).

Compositionality permits the atoms of language to locally combine with others, creating global meaning (Frege, 1948; Chomsky, 1999). As such, a complex array of meanings at the level of a phrase is explained by simple rules of composition. A natural question is whether the inherent simplicity of linguistic utterances, enabled by compositionality, manifests in representation manifolds of low complexity, described by the manifolds' intrinsic dimension. Thus far in the literature, an explicit link between degree of compositionality and representational ID has not been established. To bridge this gap, in a series of controlled experiments on causal language models and a custom dataset with tunable compositionality, we provide the first experimental insights into the relationship between the degree of compositionality of inputs and the ID of their representations over the course of training.

Using our controlled stimuli and the LMs' training data, we reproduce the established finding that LMs represent linguistic inputs on low-dimensional, nonlinear manifolds. We also show for the first time that LMs expand representations into high-dimensional linear subspaces, concretely, that **(1)** nonlinear and linear representational dimension scale differently with model size. We show the relevance

of geometry to function over LM training, in particular that **(2)** LMs' representational geometry tracks a phase transition in their linguistic competence. Different from past work, we consider two different kinds of compositionality: compositionality of *form*, or superficial combinatorial complexity, and compositionality of *meaning*, or semantic complexity; as well as two measures of dimensionality, nonlinear and linear. We not only find that geometric feature complexity reflects input compositionality, but crucially that ***nonlinear* ID encodes meaning compositionality while *linear* dimensionality encodes form compositionality**, in a way that arises over training: **(3)** nonlinear ID preserves the degree of input compositionality as an inductive bias of the model, but reflects the degree of semantic complexity at the end of training, and **(4)** linear dimensionality, not nonlinear ID, highly correlates to the superficial combinatorial complexity of inputs. Overall, results reveal a contrast between linear and nonlinear measures of feature complexity that suggests their relevance to form and meaning in how LMs process language.

## 2 BACKGROUND

**Compositionality**   It has long been a topic of debate whether neural networks also exhibit human-like compositionality when processing natural language (Fodor & Pylyshyn, 1988; Smolensky, 1990; Marcus, 2003). This debate has fueled an extensive line of empirical exploration that assesses the compositionality of neural networks in language modeling via synthetic data (Bentivogli et al., 2016; Lake & Baroni, 2018; Bahdanau et al., 2018) and natural language stimuli (Sathe et al., 2023; Dankers et al., 2022; Press et al., 2023). After the recent introduction of large language models with human-level linguistic capabilities (Wei et al., 2022), researchers have shown via mechanistic interpretability analyses that LMs often extract individual word meanings in early layers, and compose them via later-layer attention heads to construct semantic representations for multi-word expressions (Haviv et al., 2023; Geva et al., 2023). We use complementary tools to understand compositionality: rather than neurons and circuits, we link compositionality to the geometric properties of a model's embedding space which describe its learned feature complexity.

Language defines a mapping from form to meaning (de Saussure, 1916). *Form* is the physical shape of an utterance, for example, the sequence of letters or morphemes when written, or sounds when spoken. Broadly, *meaning* is the concepts or entities to which the forms refer. Unlike prior work, we make a distinction between form and meaning composition, where the formal composition relates to the combinatorial complexity of the data, and semantic composition relates to the ability to construct sentence-level meaning from word meaning. While, in grammatical sentences, meaning composition often inherits from form composition, we disentangle them by creating agrammatical versions of the dataset, further described in the Methods.

**The manifold hypothesis and low-dimensional geometry**   Deep learning problems are often considered high-dimensional, but research suggests that they have low-dimensional intrinsic structure. In computer vision, studies have shown that common learning objectives and natural image data reside on low-dimensional manifolds (Li et al., 2018; Pope et al., 2021; Valeriani et al., 2023; Psenka et al., 2024). Similarly, learning dynamics of neural LMs have been shown to occur within low-dimensional parameter subspaces (Aghajanyan et al., 2021; Zhang et al., 2023). The nonlinear, low-dimensional structure that emerges in the semantic space of these models, in contrast with models' tendency to expand representations into high-dimensional linear subspaces (Jazayeri & Ostojic, 2021), has been found to reduce learning complexity (Cheng et al., 2023; Pope et al., 2021), and likely follows from the training objective of predicting sequential observations (Recanatesi et al., 2021).

In the linguistic domain, the geometry of representations has been examined in various contexts. Recent work characterizes the organization of semantic concepts in representation space (Engels et al., 2024; Park et al., 2024; Balestriero et al., 2024; Doimo et al., 2024); it has been found that representational geometry can explicitly encode sparse tree-like syntactic structures (Andreas, 2019; Murty et al.; Alleman et al., 2021); and that linguistic categories such as part-of-speech are represented in low dimensional linear subspaces (Mamou et al., 2020; Hernandez & Andreas, 2021). Most similar to our setup, Cheng et al. (2023) reported the intrinsic dimension of representations over layers as a measure of feature complexity for several natural language datasets, finding an empirical relationship between information-theoretic and geometric compression. However, our work is the first to explicitly relate the compositionality of inputs, a critical feature of language, to the number of degrees of freedom, or intrinsic dimension, of its representation manifold.

**Language model training dynamics**   Most research on LMs focuses on the final configuration of the model at the end of pre-training. Yet, recent work shows that learning dynamics can elucidate the behavior and computational mechanisms of LMs (Chen et al., 2024; Singh et al., 2024; Tigges et al., 2024). It has been found that, over training, LMs' weight matrices become higher-rank (Abbe et al., 2023), their representations higher dimensional (Cheng et al., 2024), and their gradients increasingly diffuse (Weber et al., 2024). Over finetuning, representational dimensionality has been found to change in concert with geometric properties like cluster reorganization (Doimo et al., 2024).

Phase transitions during LM training have been found for some, but not all, aspects of language learning. Negative evidence includes that LM circuits involved in linguistic subtasks are stable (Tigges et al., 2024) and gradually reinforced (Weber et al., 2024) over training. Positive evidence for learning phase transitions includes that the ID of BERT's final [CLS] representation tracks sudden syntax acquisition and drops in training loss (Chen et al., 2024), with similar observations on Transformers trained on formal languages (Lubana et al., 2024). Our work supplements these results by investigating how the interaction between compositional understanding of language and geometric complexity of its representation arises over training.

## 3   SETUP

We consider the relationship between a dataset's degree of compositionality and its representational complexity under an LM. Here, we describe the models, dataset generation, compositionality quantification, and feature complexity estimation.

### 3.1   MODELS

We evaluate Transformer-based *causal* language models from the Pythia family (Biderman et al., 2023), as Pythia is one of the only model suites to release intermediate training checkpoints. Models are trained on The Pile, a large natural language corpus encompassing encyclopedic text, books, social media, code, and reviews (Gao et al., 2020). Over training, models are tasked to predict the next token given context, subject to a negative log-likelihood loss. Experiments are performed on all models in sizes $\in \{14m, 70m, 160m, 410m, 1.4b, 6.9b, 12b\}$.

**Pre-training analysis**   For the three intermediate sizes 410m, 1.4b, and 6.9b, we report model performance throughout the pre-training phase on the set of evaluation suites provided by (Biderman et al., 2023; Gao et al., 2024), further described in Appendix F. This encompasses a range of higher-level linguistic and reasoning tasks, spanning from long-range text comprehension (Paperno et al., 2016) to commonsense reasoning (Bisk et al., 2019). The evolution of task performance provides a cue for the type of linguistic knowledge learned by the model by a certain training checkpoint.

### 3.2   DATASETS

As we consider the relationship between the degree of compositionality and geometric feature complexity, we create a custom grammar whose compositional structure we can control. In addition, we replicate experiments on The Pile in order to compare results to a general slice of natural language.

#### 3.2.1   CONTROLLED GRAMMAR

Our stimulus dataset consists of grammatical sentences from the grammar illustrated in Figure 1. To create the grammar, we set 12 semantic categories and randomly sample a vocabulary of 50 words for each category, where the categories' vocabularies are disjoint. The categories include 5 adjective types (quality, nationality, size, color, texture), 2 noun types (job, animal) and 1 verb type. We use a simple, fixed syntactic structure by ordering the word categories:

The [quality$_1$.ADJ][nationality$_1$.ADJ][job$_1$.N]  [action$_1$.V]  the  [size$_1$.ADJ][texture.ADJ] [color.ADJ][animal.N]  then  [action$_2$.V]  the  [size$_2$.ADJ][quality$_2$.ADJ][nationality$_2$.ADJ] [job$_2$.N].

This produces sentences that are 17 words long. The order is chosen so that the generated sentences are grammatical and that the adjective order complies with the accepted ordering for English (Dixon,

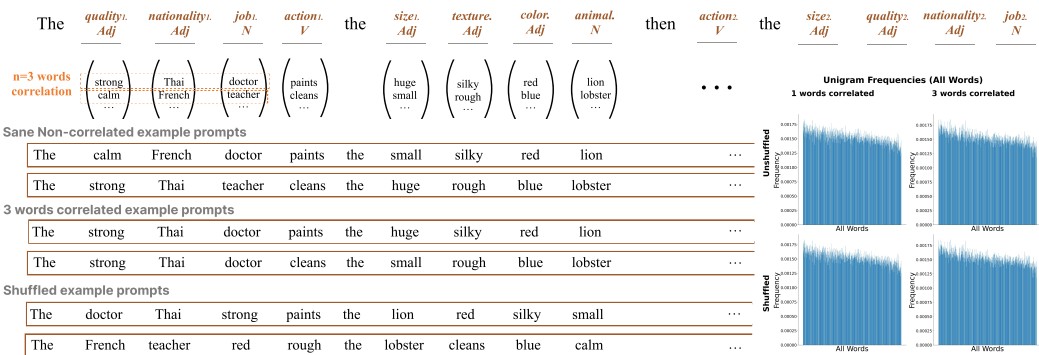

Figure 1: **Dataset structure and distributional properties. Top:** The structure of the stimulus dataset. The top row shows the ordering of word categories, such as quality.ADJ or animal.N; below it, the vocabulary for each category, including words like "strong" (quality.ADJ) and "lion" (animal.N), respectively. When controlling the degree of dataset compositionality, contiguous word positions are coupled. For instance, when $k = 3$, the first vocabulary indices for quality$_1$.ADJ, nationality$_1$.ADJ, and job$_1$.N are tied together, such that "strong Thai doctor" or "calm French teacher" can be sampled, but "strong French doctor" cannot. **Left:** Examples of generated prompts for the normal, $k = 3$, and shuffled settings. **Right:** When controlling the compositionality across $k = 1 \cdots 4$, word unigram frequencies are preserved in the resulting datasets, shown in the distributions looking identical.

1976). Vocabularies are chosen such that the sentences are semantically coherent. For example, for the first verb, the agent is a person and patient is an animal, so the possible verbs are constrained to permit "walks", but not "types". We also design grammars producing sentences of other lengths for our experiments that vary sequence length (see Appendix J. The vocabularies for each category and the structures of the different length grammars may be found in Appendix E.

Although the syntactic structure and individual vocabulary items are likely seen during training, words are sampled independently for each category without considering their probability in relationship to other words in the sentence. Therefore, generated sentences are highly unlikely to be in the training data.[1] Then, when encountering these sentences for the first time, a frozen LM must successfully construct their meanings from the meanings of their parts, or compositionally generalize.

**Controlling compositionality** We modify the grammar in order to vary the dataset's degree of compositionality. While linguistic compositionality spans many interpretations (Hupkes et al., 2019),[2] we are interested in two specific types: (1) composition of *forms*, or *combinatorial complexity* of the dataset, where a dataset is more compositional if it contains more unique word combinations; (2) composition of *meanings*, or sentence-level *compositional semantics*, where sentence meaning is composed, via syntax, from word meanings.

First, to control for dataset combinatorial complexity, we couple the values of $k$ contiguous word positions for $k = 1 \cdots 4$. That is, the sequence's atomic units are sets of $k$ adjacent words, or $k$-grams, sampled independently. This constrains the degrees of freedom in sampling to $l/k$ where $l = 12$ is the number of categories: for instance, in the 1-coupled setting, each word is sampled independently, hence 12 degrees of freedom; in the 2-coupled setting, each bigram is sampled independently, hence 6 degrees of freedom. Varying $k$ maintains the dataset's unigram distribution by design (see Figure 1 right), but constrains the dataset's $k$-gram distributions, or combinatorial complexity.

To investigate compositional semantics, we randomly shuffle the words in each sequence. This destroys syntactic coherence, and in turn, the overall meaning of the sentence. It instead preserves superficial distributional properties like word count and word co-occurrences at the sentence level,

---

[1]We cannot verify that utterances aren't in the training set, as at the time of submission, it is not possible to search The Pile.

[2]We do not consider the recursive, hierarchical nature of compositionality theorized by Chomskian linguists. We leave, e.g., different levels of syntactic embedding to future work.

as well as unigram frequencies (see Figure 1 right). Then, LM behavior on grammatically coherent vs. shuffled sequences proxies compositional vs. lexical-only semantics.

For each setting in $k \in \{1 \cdots 4\} \times \{$coherent, shuffled$\}$, we sampled a dataset of $N = 50000$ sequences, then randomly split into 5 disjoint sets of 10000 sequences. Results are reported across data splits.

**Measuring formal and semantic compositionality**    Form compositionality is controlled by the dataset combinatorial complexity. We quantify form compositionality of the controlled dataset by its Kolmogorov complexity, estimated using $\mathtt{gzip}$,[3] a popular lossless compression algorithm. We estimate the Kolmogorov complexity for $k \in \{1 \cdots 4\} \times \{$coherent, shuffled$\}$ by the $\mathtt{gzip}$-compressed dataset size in kilobytes, then correlate it to feature complexity measures (Section 3.3) for each layer.

Meaning complexity differs from form complexity. For example, the data [*cat, lion, puma*] are related semantically but not formally. As there is no unified definition for semantic complexity (Pollard & Biermann, 2000; Chersoni et al., 2016), we do not attempt to quantify it. But, as coherent sequences are grammatical and semantically coherent, it is guaranteed for coherent datasets that meaning complexity is monotonic in form complexity. In addition, as shuffling removes sequence-level semantics, meaning complexity is guaranteed to be lower on shuffled compared to coherent text, by definition.

### 3.2.2    The Pile

Although we focus on the controlled grammar in order to vary compositionality, to ensure that results are not an artifact of our prompts, we replicate experiments on The Pile, a general slice of natural language consisting of encyclopedic text, social media, reviews, news articles, and books. We uniformly sample $N = 50000$ sequences in The Pile, each consisting of 16 words, the same length as sequences in the controlled grammar, and report results over 5 random data splits.

### 3.3    Measuring feature complexity via dimensionality estimation

We are interested in how the geometric complexity of representations reflects the inputs' degree of compositionality. In particular, we consider representations in the Transformer's *residual stream* (El-hage et al., 2021). Because sequence lengths may slightly vary due to the tokenization scheme, in line with prior work (Cheng et al., 2023; Doimo et al., 2024), we aggregate over the sequence by taking the last token representation, as, due to causal attention, it is the only to attend to the entire context.

For each layer and dataset, we compute both a nonlinear and a linear measure of dimensionality. Nonlinear and linear dimensionality have key conceptual differences. The nonlinear $I_d$ is the number of degrees of freedom, or latent features, needed to describe the underlying manifold (Campadelli et al., 2015; Facco et al., 2017), see Appendix D for discussion. This differs from the *linear* effective dimensionality $d$, the dimension of the minimal linear subspace that contains the set of representations. Throughout, we will use *dimensionality* to refer to both nonlinear and linear estimates. When appropriate, we will specify $I_d$ as the nonlinear ID, $d$ as the linear effective dimensionality, and $D$ as the extrinsic dimensionality, or hidden dimension of the model. Since an $I_d$-dimensional manifold can be embedded in a $\geq I_d$-dimensional linear subspace, we always have that $I_d \leq d \leq D$.

**Intrinsic dimension**    We report the nonlinear $I_d$ using the TwoNN estimator of Facco et al., 2017. We choose TwoNN as opposed to other measures of nonlinear dimensionality for several reasons. First, it is highly correlated to other state-of-the-art estimators, such as the Maximum Likelihood Estimator (MLE) of Levina & Bickel (2004) , for both synthetic point cloud benchmarks (Facco et al., 2017) and LM representations (Cheng et al., 2023). Second, it relies on minimal assumptions of local uniformity up to the second nearest neighbor of a point, in contrast to other estimators that impose stricter assumptions, for instance, global uniformity (Albergante et al., 2019). Third, TwoNN and correlated estimators enjoy precedence in related manifold estimation literature (Cheng et al., 2023;

---

[3]The true Kolmogorov complexity is theoretically intractable. We approximate it as others have, using $\mathtt{gzip}$ (Jiang et al., 2023).

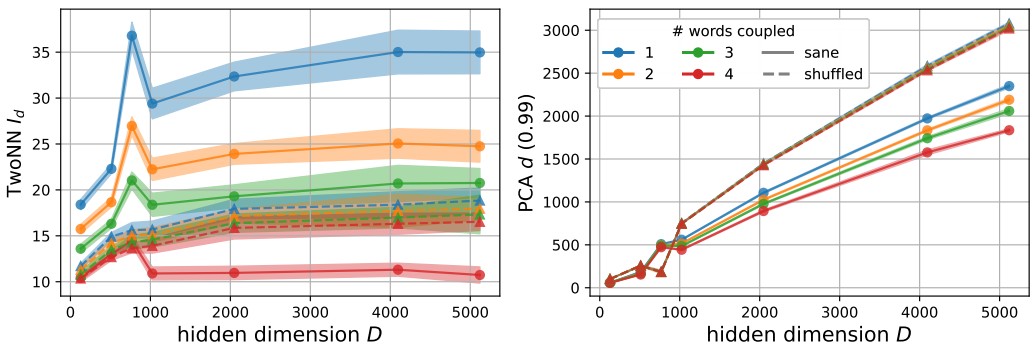

Figure 2: **Mean dimensionality over model size.** Mean nonlinear $I_d$ (left) and linear $d$ (right) over layers is shown for increasing LM hidden dimension. While nonlinear $I_d$ does not depend on hidden dimension $D$ (flat lines), PCA $d$ scales linearly in $D$. Curves are averaged over 5 data splits, $\pm$ 1 SD.

Pope et al., 2021; Chen et al., 2024; Tulchinskii et al., 2023; Ansuini et al., 2019). In addition to TwoNN in the main text, we also test MLE in Appendix C, confirming they are highly correlated.

The TwoNN estimator works as follows. Points on the underlying manifold are assumed to follow a locally homogeneous Poisson point process. Here, local refers to the neighborhood about each point $x$ encompassing $x$'s first and second nearest neighbors. Let $r_k^{(i)}$ be the Euclidean distance between $x_i$ and its $k$th nearest neighbor. Then, under the mentioned assumptions, the distance ratios $\mu_i := r_2^{(i)}/r_1^{(i)} \in [1, \infty)$ follow the cumulative distribution function $F(\mu) = (1 - \mu^{-I_d})\mathbf{1}[\mu \geq 1]$. This yields an estimator for the ID, $I_d = -\log(1 - F(\mu))/\log\mu$. Finally, given representations $\{x_i^{(j)}\}_{i=1}^N$ for LM layer $j$, $I_d^{(j)}$ is numerically fit via maximum likelihood estimation over all datapoints.

**Linear effective dimension**   To estimate the linear effective dimension $d$, we use Principal Component Analysis (PCA) (Jolliffe, 1986) with a variance cutoff of 99%. We compared to the Participation Ratio (PR) (Gao et al., 2017), a linear dimensionality measure often used in the computational neuroscience literature (cf. Chung et al. (2018); Recanatesi et al. (2019)), finding it to produce uninterpretable results, see Appendix C. For this reason, we focus on PCA in the main text.

## 4 RESULTS

We find that representational dimensionality reflects compositionality in ways that are predictable over pre-training and model scale. First, we show that language models represent linguistic data on low-dimensional, nonlinear manifolds, but in high-dimensional linear subspaces that scale linearly with the hidden dimension. Then, we show that, over training, geometric feature complexity is informative of an LM's linguistic competence, such that both exhibit a nontrivial phase transition that marks emergence of syntactic and semantic abilities. Finally, we show that representational dimensionality predictably reflects the degree of compositionality, both in terms of combinatorial complexity and sequence-level semantics and analyze its evolution over training. For brevity, we focus on model sizes 410m, 1.4b, and 6.9b in the main text, with full results in the appendix.

### 4.1 NONLINEAR AND LINEAR FEATURE COMPLEXITY SCALE DIFFERENTLY WITH MODEL SIZE

Like in previous work (Cai et al., 2021; Valeriani et al., 2023; Cheng et al., 2024), we confirm that input data are represented on a nonlinear manifold with orders-of-magnitude lower dimension than the embedding dimension. In particular, for both the controlled dataset, see Figure 2, and for The Pile, see Figure H.1, we find that $I_d \sim O(10)$ while linear $d, D \sim O(10^3)$.

Our novel finding is that nonlinear and linear dimensionality measures scale differently with model size. We fit linear regressions $D \sim \langle d \rangle_{\text{layer}}$ and $D \sim \langle I_d \rangle_{\text{layer}}$ for each setting in $k \in \{1 \cdots 4\} \times \{\text{coherent, shuffled}\}$, as well as for The Pile. Linear effect sizes $\alpha$, correlation coefficients $R$, and

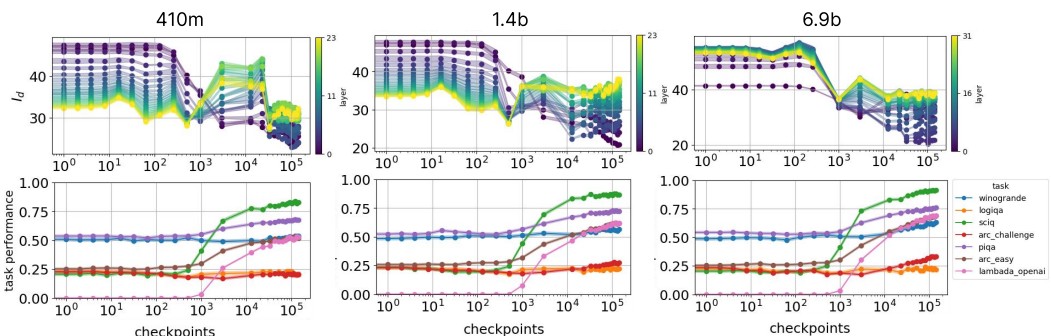

Figure 3: **ID tracks task performance**. **Top:** Layerwise $I_d$ development of Pythia-410m, 1.4b, and 6.9b over pre-training. The phase transition of ID around checkpoint $10^3$ is persistent across the model sizes. **Bottom:** Zero-shot task performance of various LM evaluation tasks of the same models across pre-training. Also around checkpoint $10^3$, linguistic competence measured by task performance starts to increase for all models.

p-values for each setting are reported in Tables G.1 and H.1 (Pile), and the curves themselves found in Figures 2 and H.1 (Pile). For the controlled dataset, $d$ scales *linearly* with hidden dimension $D$, shown in Figure 2 (right); all cases show a highly significant linear fit with $R > 0.99$ and $p < 0.005$ (Table G.1). Meanwhile, $I_d$ stabilizes to a low range $\sim O(10)$ regardless of $D$, see Figure 2 (left): here, in all cases, the effect sizes $\alpha \approx 0$ and fits are not statistically significant (Table G.1). On The Pile, Figure H.1 and Table H.1 similarly show that $d \propto D$, where the linear relationship is highly significant; the high effect size $\alpha = 0.81$, in this case, indicates that the model tends to fill the ambient space such that $d \approx D$. While for The Pile, $I_d \propto D$ ($R = 0.95$, $p < 0.001$) as well, the tiny effect size $\alpha = 0.002$ shows that $I_d$ changes negligibly with respect to $D$, seen in Figure H.1 (left).

These results highlight key differences in how linear and nonlinear dimensions are recruited: LMs *globally* distribute representations to occupy $d \propto D$ dimensions of the space, but their shape is *locally* constrained to a low-dimensional ($I_d$) manifold. Robustness of $I_d$ to scaling the hidden dimension suggests that LMs, once sufficiently performant, recover the degrees of freedom underlying the data.

### 4.2 EVOLUTION OF REPRESENTATIONAL GEOMETRY TRACKS EMERGENT LINGUISTIC ABILITIES OVER TRAINING

We just saw how dimensionality scales with size, and now we investigate its change over time. We find that feature complexity is highly related to the LM's linguistic capabilities, assessed using the eval-harness benchmark performance, over training. Figure 3 shows the evolution of $I_d$ on the $k = 1$ dataset (top), where each curve is one layer, with the evolution of LM performance on the benchmark tasks (bottom), where each curve plots performance on an individual task.

We observe in Figure 3 that, for all models, the evolution of representational dimensionality closely tracks a sudden transition in LM task performance. In Figure 3 top, we first observe that $I_d$ decreases sharply before checkpoint $10^3$ and then re-distributes. At the same time, task performance rapidly improves after the steep decrease in $I_d$ (Figure 3 bottom). Feature complexity evolution on The Pile is shown in Figure H.2, and exhibits a similar transition to that reported in Figure 3. Further, the existence of the phase transition in representational geometry $t \approx 10^3$ is robust to the dimensionality measure and whether the data are shuffled, see Figure G.5. Our results resonate with Chen et al. (2024), who observed in BERT models a similar two-part $I_d$ transition on the training corpus; they showed that the two extrema corresponding to the dip and uptick in $I_d$ temporally coincided with the onset of higher-order linguistic capabilities. Together, results show that representational complexity can signify whether and when LMs learn linguistic structure. Crucially, we show that the phase transition exists for inputs *beyond* in-distribution data, which was the subject of (Chen et al., 2024), and, furthermore, beyond grammatical data (Figure G.5) as a more general property of LM processing.

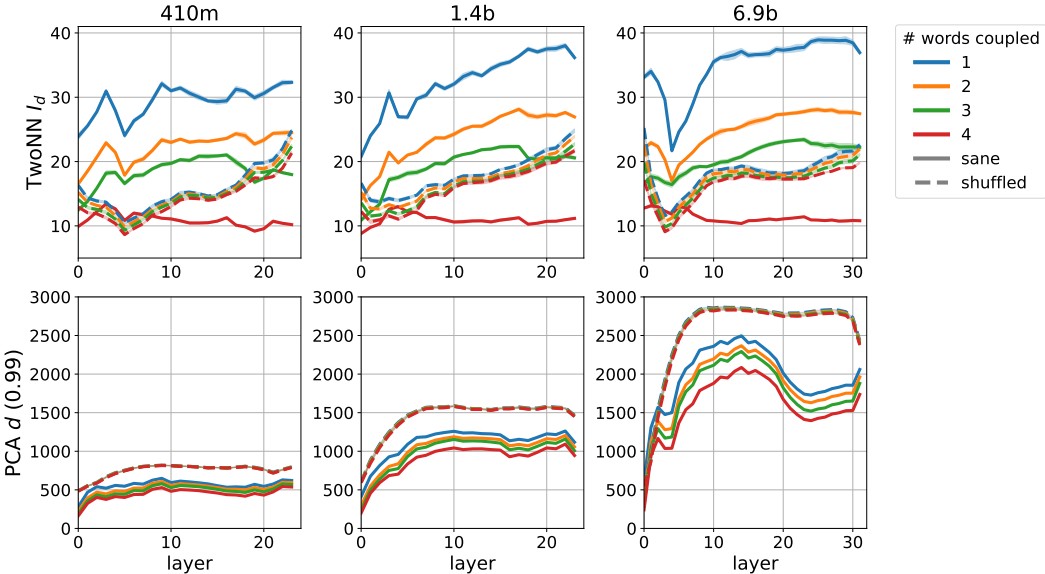

Figure 4: **Dimensionality over layers.** Nonlinear $I_d$ (top) and linear $d$ (bottom) over layers are shown for sizes 410m, 1.4b, and 6.9b (left to right). Each color corresponds to a coupling length $k \in 1 \cdots 4$. Solid curves denote coherent sequences, and dotted curves denote shuffled sequences. For all models, lower $k$ results in higher $I_d$ and $d$ for both normal and shuffled settings. For all models, shuffling results in lower $I_d$ but higher $d$. Curves are averaged over 5 random data splits, shown with $\pm 1$ SD (shaded); SDs across random data splits tended to be very small.

### 4.3 REPRESENTATIONAL COMPLEXITY REFLECTS INPUT COMPOSITIONALITY

We just established that feature complexity is informative of when models gain complex linguistic capabilities that, by definition, require compositional understanding. Now, we establish our key result, which is that **feature complexity encodes input compositionality**, both when considering formal compositionality, or data combinatorial complexity, as well as meaning compositionality, or sentence-level semantics. We first show that this holds for fully-trained models that have reached final linguistic competency. Then, using evidence from the training phase of the model, we show that the correspondence between feature complexity and input compositionality is present first as an inductive bias of the model that encodes formal complexity; but then, that it persists due to learned syntactic and semantic features that encode meaning complexity. Lastly, we further develop the coding differences between $d$ and $I_d$, confirming an existing hypothesis in the literature (Recanatesi et al., 2021) that they respectively encode formal and semantic complexity of inputs.

**Data combinatorial complexity**   On fully-trained models, representational dimensionality preserves relative dataset compositionality. Figure 4 shows for fully-trained Pythia 410m, 1.4b, and 6.9b that $I_d$ and $d$ increase with the degree of formal compositionality within both coherent (solid) and shuffled (dashed) settings: the highest curves (blue) correspond to the $k = 1$ dataset, or 12 degrees of freedom, and the lowest (red) denote the $k = 4$ dataset, or 3 degrees of freedom. The relative order of feature complexity, moreover, holds for all layers, seen by non-overlapping solid curves in Figure 4.

Grammaticality is not a precondition for representational dimensionality to reflect data combinatorial complexity: in Figure 4 (top), dashed curves corresponding to shuffled text are also ordered $k = 1 \cdots 4$ top to bottom. While the relative order of formal complexity is preserved in the LM's feature complexity for both grammatical and agrammatical datasets, the separation is greater for grammatical text (solid curves). We hypothesize that this is due to shuffled text being out-of-distribution, such that the model cannot integrate the sequences' meaning, but nevertheless preserve surface-level complexity in its representations. This tendency holds for pre-trained models of all sizes (see Figure G.1) and for sentences of different lengths (see Figure J.2).

The relationship between dimensionality and data combinatorial complexity, controlled by $k$, for coherent text is *not* an emergent feature over training. In Figure 5 (left), the inverse relationship

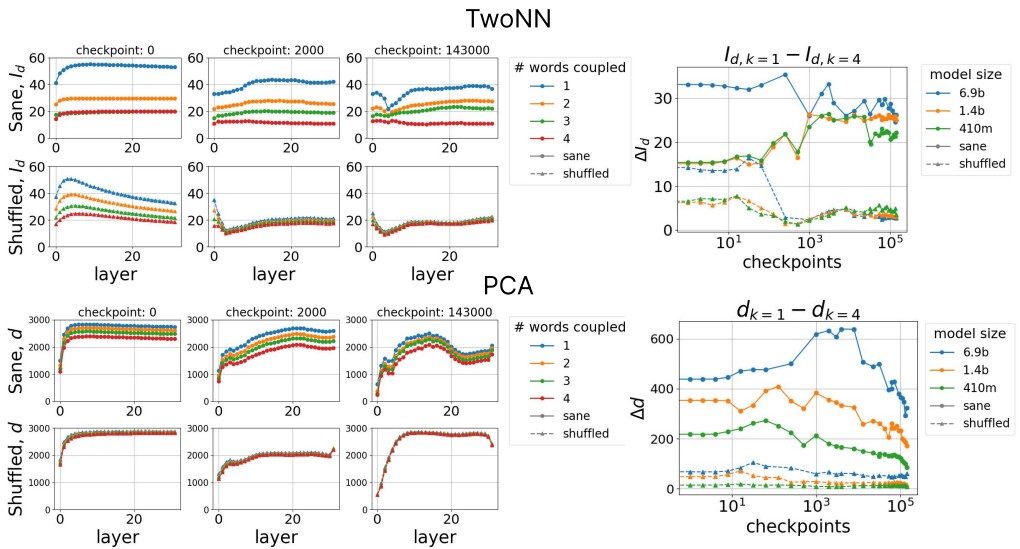

Figure 5: **Training dynamics of dimensionality.** The top row shows results for TwoNN $I_d$ and the bottom row shows the PCA $d$. **(Left):** Layerwise $I_d$ at different timepoints of training for coherent vs. shuffled examples with different coupling $k$ (6.9b model). $I_d$ difference of shuffled examples with varying $k$ diminishes as the training persists. All curves are shown with $\pm 1$ SD (the SDs were very small). **(Right):** $\Delta I_d$ between $k = 1, k = 4$ across training for various model sizes (different colors).

between $k$ and both $I_d$ and $d$ is present throughout training. But, the reason for this relation differs at the start and end: in shuffled text, where sequence-level semantics are not present, the relationship between $k$ and dimensionality is salient at the *beginning* and greatly diminishes by the end, whereas in coherent text it remains salient throughout training. Together, these demonstrate an inductive bias of the initialized LM architecture to preserve input complexity in its representations. Then, over training, differences in dimensionality may be increasingly explained by features beyond the surface distribution of inputs. We claim that these features are semantic, providing evidence towards the claim in what follows.

**Compositional semantics** Shuffling sequences destroys their meaning, removing dataset complexity attributed to sentence-level semantics. Figure 4 shows, for fully-trained models, the dimensionality over layers for coherent and shuffled inputs. Here, nonlinear and linear dimensionalities show opposing trends: $I_d$ for shuffled text *collapses* to a low range, while $d$ *increases*, seen by the dashed curves in each plot compared to solid curves. Furthermore, this tendency holds across all model sizes (see Figure G.1).

We refer to the phenomenon in which shuffling destroys sequence-level semantics and $I_d$, also attested for sequences in Pythia's training corpus (Cheng et al., 2024), as *shuffling feature collapse*. Evidence from the training dynamics of the LM further suggests that this feature collapse is due to semantics. We saw in Section 4.2 that training step $t = 10^3$ approximately marked a phase transition after which the LM's linguistic competencies sharply rose. Crucially, the epoch $t = 10^3$ preceding the sharp increase in linguistic capabilities is also the first to exhibit shuffling feature collapse. Figure 5 (right) shows the $\Delta I_d$ between the $k = 1$ and $k = 4$ dataset for several model sizes, across training (x-axis). Shuffling feature collapse, given by low $\Delta I_d$, occurs around $t = 10^3$ for all models. On the other hand, $\Delta I_d$ for coherent text stabilizes to around $\sim 25$ for different model sizes. This transition does not occur for linear $d$, see Figure 5 (right, bottom). This suggests that shuffling feature collapse for $I_d$ is symptomatic of when the LM learns to extract meaningful semantic features.

We interpret shuffling feature collapse using an argument from Recanatesi et al. (2021), who propose that predictive coding requires the model to satisfy two objectives: to encode the vast "space of inputs and outputs", exerting upward pressure on representational complexity, and at the same time, to extract latent features to support prediction, exerting a downward pressure on complexity. Recanatesi et al. (2021) suggest that the first pressure expands the linear representation space $\mathbb{R}^d$, while the

Table 1: //++ LLAMA and MISTRAL// **Spearman correlations between dimensionality and estimated Kolmogorov complexity**. The Spearman correlation $\rho$ between the `gzip`ped dataset size (KB) and representational dimensionality (rows), averaged over layers, is shown for all tested **Pythia** model sizes (left 7 columns) as well as **Llama** and **Mistral** (right 2 columns). Values marked with a * are significant with a p-value threshold of 0.05. Values marked with † are significant with a p-value threshold of 0.1. Across models, average-layer $I_d$ is not correlated to the estimated Kolmogorov complexity, or formal compositionality, of datasets. Average-layer linear $d$ is consistently highly positively correlated to the estimated Kolmogorov complexity, except one outlier (160m).

| Spearman $\rho$ | 14m | 70m | 160m | 410m | 1.4b | 6.9b | 12b | Llama | Mistral |
|---|---|---|---|---|---|---|---|---|---|
| $I_d$ | -0.20 | -0.06 | -0.20 | -0.05 | 0.04 | 0.01 | 0.05 | -0.36 | 0.00 |
| $d$ | 0.90* | 0.47† | -0.50† | 0.96* | 0.96* | 0.92* | 0.86* | 1.0* | 1.0* |

second compresses representations to a $I_d$-dimensional manifold. Indeed, in our setting, shuffling words increases input complexity, thus increasing $d$. But, shuffling destroys sequence semantics, exerting a downward pressure on $I_d$. Recanatesi et al. (2021)'s interpretation of linear dimensions as encoding the input space corresponds to what we have been calling *formal compositionality*; conversely, what they refer to as latent semantic features, encoded nonlinearly, is aligned with our *meaning compositionality*. We now investigate this form-meaning coding dichotomy in more detail.

**Form-meaning dichotomy in representation learning**  We proposed in line with Recanatesi et al. (2021) that linear $d$ captures surface-level variation, while $I_d$ primarily encodes semantic variation. We have shown the latter in the previous section: $I_d$ decreases in the absence of compositional semantics while $d$ does not, suggesting that $I_d$, not $d$, encodes sequence-level meaning complexity. If this hypothesis holds, we need to show that linear $d$, and not $I_d$, encodes form compositionality.

Form compositionality is quantified by the `gzip`-compressed size of each dataset. Spearman correlations between `gzip` (kilobytes), and dimensionality are shown in Table 1. Consistently across model sizes and families, average layerwise $I_d$ is not correlated to `gzip` size, while average layerwise $d$ is highly correlated to `gzip` size; we discuss the outlier 160m in Appendix I. The high correlation between $d$ and `gzip` size is, moreover, surprisingly consistent across layers, see Figures I.2 and I.3, and already present as an inductive bias of the initialized model (see Figure I.5, Appendix I for training dynamics discussion), while the correlation to $I_d$ is highly variable and seldom significant for all of training. This suggests form complexity, already present in the inputs to the LM, is superficially *preserved*, while meaning complexity is instead *constructed*, over layers and over training.

## 5 DISCUSSION

We have studied language model compositionality from a geometric and dynamic perspective. Using a carefully curated synthetic dataset, we found strong relationships between the compositionality of linguistic expressions and the dimensionality of their representations. On one hand, representational dimensionality is positively correlated to datasets' formal, or combinatorial, complexity. On the other hand, grammatical sequences, whose semantics are composed via syntax, tend to exhibit a higher non-linear dimensionality, but a lower linear dimensionality, than agrammatical shuffled sequences. We showed that the positive relationship between compositionality and dimensionality is an inductive bias of the model, but that it is eventually shed in favor of learning a representation manifold that reflects meaningful *semantic* complexity in a phase transition. Results suggested differential coding of form and meaning in LM representations, where form complexity, estimated with `gzip`, is expressed linearly, and meaning complexity is expressed nonlinearly.

A central throughline in our results is that LMs compress representations to low-dimensional nonlinear manifolds, yet expand them to high-dimensional linear subspaces. This echoes independent results in computational neuroscience by Manley et al. (2024), who found that linear dimensionality scaled with number of neurons recorded in the mouse cortex, and De & Chaudhuri (2023), who found in neural networks that nonlinear, rather than linear, dimensionality better captured task semantics. The tendency for LMs to compress data to low-dimensional nonlinear manifolds, but, at the same time,

expand them into high-dimensional linear subspaces, suggests a solution to the curse of dimensionality that also enjoys its blessings. High-dimensional representations classically engender overfitting and poor generalization (Hughes, 1968); but, these high-dimensional representations may lie on manifolds whose ID actually captures the latent sparsity of the data (De & Chaudhuri, 2023). At the same time, more dimensions implies more expressive orthogonality relations and linear decodability of categories (Cohen et al., 2020; Elmoznino & Bonner, 2023; Sorscher et al., 2022). Benefits of this dual patterning of intrinsic and effective dimensionality have been observed in biological and artificial intelligence (Jazayeri & Ostojic, 2021; Recanatesi et al., 2021; Haxby et al., 2011; Huth et al., 2012; De & Chaudhuri, 2023), where moreover, (linear) dimensional expansion and compression have implications for "lazy" and "rich" feature learning regimes, respectively (Flesch et al., 2022). While the present work is the first to show that this dual patterning in LMs broadly corresponds to a form-meaning dichotomy in representation learning, further work is needed to distentangle how nonlinear and linear features causally contribute to predictive coding.

## REPRODUCIBILITY STATEMENT

In Section 3.1, we have outlined the language models used in our experiments. The synthetic and naturalistic datasets employed to study compositionality and geometric feature complexity are introduced in Section 3.2 and further detailed in Appendix E. A comprehensive description of the measures used to assess representation geometric complexity is provided in Section 3.3, Appendix D, and Appendix C. Additionally, the benchmark tasks used to evaluate the Pythia checkpoints are summarized in Appendix F. Computing resources are described in Appendix A, and links to the assets used and their licenses are provided in Appendix B.

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

## A    COMPUTING RESOURCES

All experiments were run on a cluster with 12 nodes with 5 NVIDIA A30 GPUs and 48 CPUs each. Extracting LM representations took a few wall-clock hours per model-dataset computation. ID computation took approximately 0.5 hours per model-dataset computation. Taking parallelization into account, we estimate the overall wall-clock time taken by all experiments, including failed runs, preliminary experiments, etc., to be of about 10 days.

## B    ASSETS

**Pile** https://huggingface.co/datasets/NeelNanda/pile-10k;         license: bigscience-bloom-rail-1.0

**Pythia** https://huggingface.co/EleutherAI/pythia-6.9b-deduped;   license: apache-2.0

**scikit-dimension** https://scikit-dimension.readthedocs.io/en/latest/;   license: bsd-3-clause

**PyTorch** https://scikit-learn.org/; license: bsd

## C    OTHER DIMENSIONALITY ESTIMATORS

**Maximum Likelihood Estimator**    In addition to TwoNN, we considered Levina & Bickel (2004)'s Maximum Likelihood Estimator (MLE), a similar, nonlinear measure of $I_d$. MLE has been used in prior works on representational geometry such as (Cai et al., 2021; Cheng et al., 2023; Pope et al., 2021), and similarly models the number of points in a neighborhood around a reference point $x$ to follow a Poisson point process. For details we refer to the original paper (Levina & Bickel, 2004). Like past work (Facco et al., 2017; Cheng et al., 2023), we found MLE and TwoNN to be highly correlated, producing results that were nearly identical: compare Figure 2 left to Figure G.4 left, and Figure G.1 top to Figure G.3 top).

**Participation Ratio**    For our primary linear measure of dimensionality $d$, we computed PCA and took the number of components that explain 99% of the variance. In addition to PCA, we computed the Participation Ratio (PR), defined as $(\sum_i \lambda_i)^2 / (\sum_i \lambda_i^2)$ (Gao et al., 2017). We found PR to give results that were incongruous with intuitions about linear dimensionality. In particular, it produced a lower dimensionality estimate than the nonlinear estimators we tested; see, e.g., Figure G.4, where the PR-$d$ for coherent text is less than that of TwoNN. This contradicts the mathematical relationship that $I_d \leq d \leq D$. This may be because, empirically, PR-$d$ corresponded to explained variances of $60 - 80\%$, which are inadequate to describe the bounding linear subspace for the representation manifold. Therefore, while we report the mean PR-$d$ over model size in Figure G.4 and the dimensionality over layers in Figure G.3 for completeness, we do not attempt to interpret them.

## D    INTRINSIC DIMENSION DETAILS

ID estimation methods practically rely on a finite set of points and their nearest-neighbor structure in order to compute an estimated dimensionality value. The underlying geometric calculations assume

that these are points sampled from a continuum, such as a lower-dimensional non-linear manifold. In our case, we actually have a discrete set of points so the notion of an underlying manifold is not strictly applicable. However, we can ask the question: if those points had been sampled from a manifold, what would the estimated ID be? Since the algorithms themselves only require a discrete set of points, they can be used to answer that question.

# E  CONTROLLED GRAMMAR

We design 5 different grammars of varying lengths (5, 8, 11, 15, and 17 words). The 17 word grammar is the one used for all controlled grammar experiments except the "Varying Sequence Length" experiments (appendix J). The structures of the grammars can be found below.

## E.1  LENGTH: 5 WORDS

The [$job_1$.N] [$action_1$.V] the [animal.N].

## E.2  LENGTH: 8 WORDS

The [$nationality_1$.ADJ][$job_1$.N] [$action_1$.V] the [color.ADJ][texture.ADJ] [animal.N]

## E.3  LENGTH: 11 WORDS

The  [$size_2$.ADJ][$quality_1$.ADJ][$nationality_1$.ADJ][$job_1$.N]  [$action_1$.V]  the  [$size_1$.ADJ] [color.ADJ] [texture.ADJ] [animal.N]

## E.4  LENGTH: 15 WORDS

The  [$quality_1$.ADJ][$nationality_1$.ADJ][$job_1$.N]  [$action_1$.V]  the [$size_1$.ADJ][color.ADJ][texture.ADJ][animal.N] then [$action_2$.V] the [$size_2$.ADJ][$job_2$.N].

## E.5  LENGTH: 17 WORDS

The  [$quality_1$.ADJ][$nationality_1$.ADJ][$job_1$.N]  [$action_1$.V]  the  [$size_1$.ADJ][texture.ADJ] [color.ADJ][animal.N]  then  [$action_2$.V]  the  [$size_2$.ADJ][$quality_2$.ADJ][$nationality_2$.ADJ] [$job_2$.N].

Each category, colored and enclosed in brackets, is sampled from a vocabulary of 50 possible words, listed in the table below:

| Category | Words |
| --- | --- |
| $job_1$ | teacher, doctor, engineer, chef, lawyer, plumber, electrician, accountant, nurse, mechanic, architect, dentist, programmer, photographer, painter, firefighter, police, pilot, farmer, waiter, scientist, actor, musician, writer, athlete, designer, carpenter, librarian, journalist, psychologist, gardener, baker, butcher, tailor, cashier, barber, janitor, receptionist, salesperson, manager, tutor, coach, translator, veterinarian, pharmacist, therapist, driver, bartender, security, clerk |

| | |
|---|---|
| job$_2$ | banker, realtor, consultant, therapist, optometrist, astronomer, biologist, geologist, archaeologist, anthropologist, economist, sociologist, historian, philosopher, linguist, meteorologist, zoologist, botanist, chemist, physicist, mathematician, statistician, surveyor, pilot, steward, dispatcher, ichthyologist, oceanographer, ecologist, geneticist, microbiologist, neurologist, cardiologist, pediatrician, surgeon, anesthesiologist, radiologist, dermatologist, gynecologist, urologist, psychiatrist, physiotherapist, chiropractor, nutritionist, personal trainer, yoga instructor, masseur, acupuncturist, paramedic, midwife |
| animal | dog, cat, elephant, lion, tiger, giraffe, zebra, monkey, gorilla, chimpanzee, bear, wolf, fox, deer, moose, rabbit, squirrel, raccoon, beaver, otter, penguin, eagle, hawk, owl, parrot, flamingo, ostrich, peacock, swan, duck, frog, toad, snake, lizard, turtle, crocodile, alligator, shark, whale, dolphin, octopus, jellyfish, starfish, crab, lobster, butterfly, bee, ant, spider, scorpion |
| color | red, blue, green, yellow, purple, orange, pink, brown, gray, black, white, cyan, magenta, turquoise, indigo, violet, maroon, navy, olive, teal, lime, aqua, coral, crimson, fuchsia, gold, silver, bronze, beige, tan, khaki, lavender, plum, periwinkle, mauve, chartreuse, azure, mint, sage, ivory, salmon, peach, apricot, mustard, rust, burgundy, mahogany, chestnut, sienna, ochre |
| size$_1$ | big, small, large, tiny, huge, giant, massive, microscopic, enormous, colossal, miniature, petite, compact, spacious, vast, wide, narrow, slim, thick, thin, broad, expansive, extensive, substantial, boundless, considerable, immense, mammoth, towering, titanic, gargantuan, diminutive, minuscule, minute, hulking, bulky, hefty, voluminous, capacious, roomy, cramped, confined, restricted, limited, oversized, undersized, full, empty, half, partial |
| size$_2$ | lengthy, short, tall, long, deep, shallow, high, low, medium, average, moderate, middling, intermediate, standard, regular, normal, ordinary, sizable, generous, abundant, plentiful, copious, meager, scanty, skimpy, inadequate, sufficient, ample, excessive, extravagant, exorbitant, modest, humble, grand, majestic, imposing, commanding, dwarfed, diminished, reduced, enlarged, magnified, amplified, expanded, contracted, shrunken, swollen, bloated, inflated, deflated |
| nationality$_1$ | American, British, Canadian, Australian, German, French, Italian, Spanish, Japanese, Chinese, Indian, Russian, Brazilian, Mexican, Argentinian, Turkish, Egyptian, Nigerian, Kenyan, African, Swedish, Norwegian, Danish, Finnish, Icelandic, Dutch, Belgian, Swiss, Austrian, Greek, Polish, Hungarian, Czech, Slovak, Romanian, Bulgarian, Serbian, Croatian, Slovenian, Ukrainian, Belarusian, Estonian, Latvian, Lithuanian, Irish, Scottish, Welsh, Portuguese, Moroccan, Algerian |
| nationality$_2$ | Vietnamese, Thai, Malaysian, Indonesian, Filipino, Singaporean, Nepalese, Bangladeshi, Maldivian, Pakistani, Afghan, Iranian, Iraqi, Syrian, Lebanese, Israeli, Saudi, Emirati, Qatari, Kuwaiti, Omani, Yemeni, Jordanian, Palestinian, Bahraini, Tunisian, Libyan, Sudanese, Ethiopian, Somali, Ghanaian, Ivorian, Senegalese, Malian, Cameroonian, Congolese, Ugandan, Rwandan, Tanzanian, Mozambican, Zambian, Zimbabwean, Namibian, Botswanan, New Zealander, Fijian, Samoan, Tongan, Papuan, Marshallese |

| action$_1$ | feeds, walks, grooms, pets, trains, rides, tames, leashes, bathes, brushes, adopts, rescues, shelters, houses, cages, releases, frees, observes, studies, examines, photographs, films, sketches, paints, draws, catches, hunts, traps, chases, pursues, tracks, follows, herds, corrals, milks, shears, breeds, mates, clones, dissects, stuffs, mounts, taxidermies, domesticates, harnesses, saddles, muzzles, tags, chips, vaccinates |
|---|---|
| action$_2$ | hugs, kisses, loves, hates, admires, respects, befriends, distrusts, helps, hurts, teaches, learns from, mentors, guides, counsels, advises, supports, undermines, praises, criticizes, compliments, insults, congratulates, consoles, comforts, irritates, annoys, amuses, entertains, bores, inspires, motivates, discourages, intimidates, impresses, disappoints, surprises, shocks, delights, disgusts, forgives, resents, envies, pities, understands, misunderstands, trusts, mistrusts, betrays, protects |
| quality$_1$ | good, bad, excellent, poor, superior, inferior, outstanding, mediocre, exceptional, sublime, superb, terrible, wonderful, awful, great, horrible, fantastic, dreadful, marvelous, atrocious, splendid, appalling, brilliant, dismal, fabulous, lousy, terrific, abysmal, incredible, substandard, amazing, disappointing, extraordinary, stellar, remarkable, unremarkable, impressive, unimpressive, admirable, despicable, praiseworthy, blameworthy, commendable, reprehensible, exemplary, subpar, ideal, flawed, perfect, imperfect |
| quality$_2$ | acceptable, unacceptable, satisfactory, unsatisfactory, sophisticated, insufficient, adequate, exquisite, suitable, unsuitable, appropriate, inappropriate, fitting, unfitting, proper, improper, correct, incorrect, right, wrong, accurate, inaccurate, precise, imprecise, exact, inexact, flawless, faulty, sound, unsound, reliable, unreliable, dependable, undependable, trustworthy, untrustworthy, authentic, fake, genuine, counterfeit, legitimate, illegitimate, valid, invalid, legal, illegal, ethical, unethical, moral, immoral |
| texture | smooth, rough, soft, hard, silky, coarse, fluffy, fuzzy, furry, hairy, bumpy, lumpy, grainy, gritty, sandy, slimy, slippery, sticky, tacky, greasy, oily, waxy, velvety, leathery, rubbery, spongy, springy, elastic, pliable, flexible, rigid, stiff, brittle, crumbly, flaky, crispy, crunchy, chewy, stringy, fibrous, porous, dense, heavy, light, airy, feathery, downy, woolly, nubby, textured |

## F  BENCHMARK TASKS

Here we briefly summarize the benchmark tasks that we use to evaluate Pythia checkpoints as described in Section 4.3. In figure 3, we did not include WSC (Winogrande Schema Challenge) which was originally included in Biderman et al., as it has been proposed that WSC dataset performance on LMs might be corrupted by spurious biases in the dataset (Sakaguchi et al., 2021). Instead, we only presented the evaluation from WinoGrande task, which is inspired from original WSC task but adjusted to reduce the systematic bias (Sakaguchi et al., 2021).

**WinoGrande**  WinoGrande (Sakaguchi et al., 2021) is a dataset designed to test commonsense reasoning by building on the structure of the Winograd Schema Challenge (Levesque et al., 2012). It presents sentence pairs with subtle ambiguities where understanding the correct answer requires world knowledge and commonsense reasoning. It challenges models to differentiate between two possible resolutions of pronouns or references, making it a benchmark for evaluating an AI's ability to understand context and reasoning.

**LogiQA**    LogiQA (Liu et al., 2020) is an NLP benchmark for evaluating logical reasoning abilities in models. It consists of multiple-choice questions derived from logical reasoning exams for human students. The questions test various forms of logical reasoning, such as deduction, analogy, and quantitative reasoning, making it ideal for assessing how well AI can handle structured logical problems.

**SciQ**    SciQ (Welbl et al., 2017) is a dataset focused on scientific question answering, based on material from science textbooks. It features multiple-choice questions related to science topics like biology, chemistry, and physics. The benchmark is designed to test a model's ability to comprehend scientific information and answer questions using factual knowledge and reasoning.

**ARC Challenge**    The ARC (AI2 Reasoning Challenge) Challenge Set (Clark et al., 2018) is a benchmark designed to test models on difficult, grade-school-level science questions. It presents multiple-choice questions that are challenging due to requiring complex reasoning, inference, and background knowledge beyond simple retrieval-based approaches. It is a tougher subset of the larger ARC dataset.

**PIQA**    PIQA (Physical Interaction QA) (Bisk et al., 2020) is a benchmark designed to test models on physical commonsense reasoning. The questions require understanding basic physical interactions, like how objects interact or how everyday tasks are performed. It focuses on scenarios that involve intuitive knowledge of the physical world, making it a useful benchmark for evaluating practical commonsense in models.

**ARC Easy**    ARC Easy is the easier subset of the AI2 Reasoning Challenge, consisting of grade-school-level science questions that require less complex reasoning compared to the Challenge set. This benchmark is meant to evaluate models' ability to handle straightforward factual and retrieval-based questions, making it more accessible for baseline NLP models.

**LAMBADA**    LAMBADA (Paperno et al., 2016) is a reading comprehension benchmark where models must predict the last word of a passage. The challenge lies in the fact that understanding the entire context of the passage is necessary to guess the correct word. This benchmark tests a model's long-range context comprehension and coherence skills in natural language.

# G    ADDITIONAL RESULTS: CONTROLLED GRAMMAR

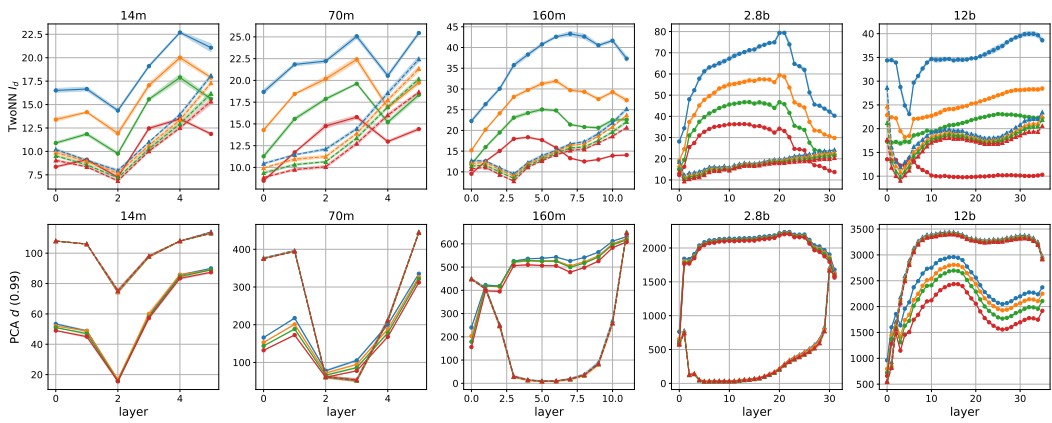

Figure G.1: **Dimensionality over layers.** TwoNN nonlinear $I_d$ (top) and PCA linear $d$ (bottom) over layers are shown for all sizes (left to right). Each color corresponds to a coupling length $k \in 1 \cdots 4$. Solid curves denote coherent sequences, and dotted curves denote shuffled sequences. For all models, lower $k$ results in higher $I_d$ and $d$ for both normal and shuffled settings. For all models, shuffling results in lower $I_d$ but higher $d$. Curves are averaged over 5 random seeds, shown with $\pm 1$ SD.

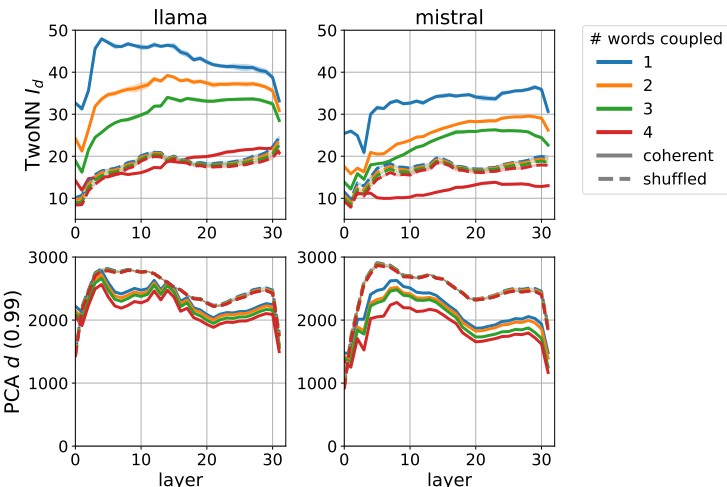

Figure G.2: //++ NEW// **Dimensionality over layers for Llama-3-8B and Mistral-7B.** TwoNN nonlinear $I_d$ (top) and PCA linear $d$ (bottom) over layers are shown for all sizes (left to right). Each color corresponds to a coupling length $k \in 1 \cdots 4$. Solid curves denote coherent sequences, and dotted curves denote shuffled sequences. For all models, lower $k$ results in higher $I_d$ and $d$ for both normal and shuffled settings. For all models, shuffling results in lower $I_d$ but higher $d$. Curves are averaged over 5 random seeds, shown with $\pm 1$ SD. **Results mirror those of the Pythia models.**

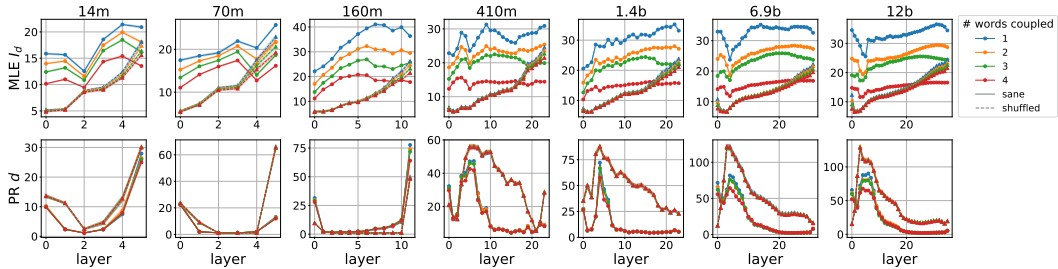

Figure G.3: **Other dimensionality metrics over layers.** MLE nonlinear $I_d$ (top) and PR linear $d$ (bottom) over layers are shown for all model sizes (left to right). Each color corresponds to a coupling length $k \in 1 \cdots 4$. Solid curves denote coherent sequences, and dotted curves denote shuffled sequences. For all models, lower $k$ results in higher $I_d$ for both normal and shuffled settings. For all models, shuffling results in lower $I_d$. The PR-$d$ produced nonsensical results, with linear dimensionality higher than nonlinear dimensionality. Curves are averaged over 5 random seeds, shown with $\pm 1$ SD.

| Mode | k-coupling | PCA $d$ | | | TwoNN $I_d$ | | |
|------|-----------|---------|---|---|-------------|---|---|
| | | $\alpha$ | R | p-value | $\alpha$ | R | p-value |
| coherent | 1 | 0.4598 | 0.9956 | $2 \times 10^{-6}$ | 0.0023 | 0.6341 | 0.1261 |
| coherent | 2 | 0.4268 | 0.9954 | $3 \times 10^{-6}$ | 0.0011 | 0.5580 | 0.1930 |
| coherent | 3 | 0.4014 | 0.9943 | $5 \times 10^{-6}$ | 0.0009 | 0.6616 | 0.1056 |
| coherent | 4 | 0.3569 | 0.9924 | $1 \times 10^{-5}$ | -0.0003 | -0.3523 | 0.4383 |
| shuffled | 1 | 0.6239 | 0.9919 | $1.1 \times 10^{-5}$ | 0.0011 | 0.8488 | 0.0157 |
| shuffled | 2 | 0.6193 | 0.9917 | $1.2 \times 10^{-5}$ | 0.0010 | 0.8487 | 0.0157 |
| shuffled | 3 | 0.6153 | 0.9916 | $1.2 \times 10^{-5}$ | 0.0010 | 0.8586 | 0.0134 |
| shuffled | 4 | 0.6114 | 0.9916 | $1.2 \times 10^{-5}$ | 0.0009 | 0.8559 | 0.0140 |

Table G.1: **Linear regression of average layerwise dimensionality to hidden dimension,** $D$. For each setting (Mode, $k$-coupling columns) and dimensionality measure (PCA, TwoNN columns), the linear effect size $\alpha$ along with $R$-value and $p$-value are reported. PCA linear dimension shows a consistent strong linear relationship with large effect size $\alpha$ to hidden dimension $D$ ($p < 0.001$) for all settings in $k = \{1 \cdots 4\} \times [\text{coherent, shuffled}]$. TwoNN intrinsic dimension does not scale linearly as $D$ in all settings, showing a non-significant relationship for coherent text and a significant one for shuffled text. For all TwoNN settings, the effect size $\alpha$ is near-zero, showing that nonlinear $I_d$ is robust to changes in hidden dimension $D$.

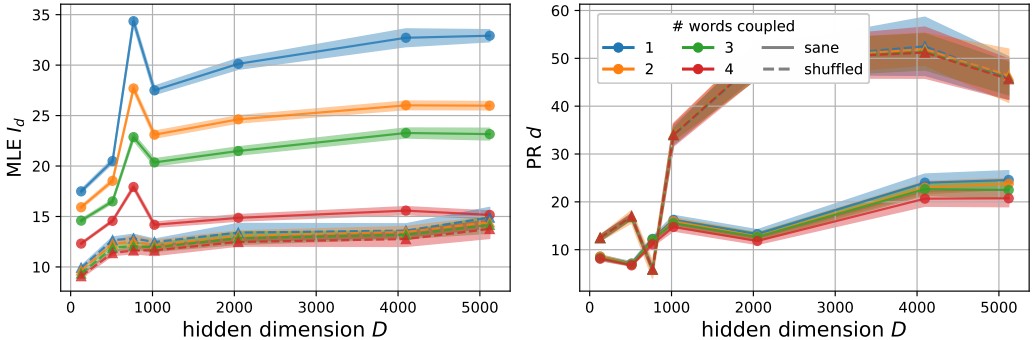

Figure G.4: **Mean dimensionality over model size (other metrics).** Mean nonlinear $I_d$ computed with MLE (left) and linear $d$ computed with PR (right) over layers is shown for increasing LM hidden dimension $D$. MLE $I_d$ does not depend on extrinsic dimension $D$ (flat lines). PR $d$ produces nonsensical values, higher than the nonlinear $I_d$. Curves are averaged over 5 random seeds, shown with $\pm$ 1 SD.

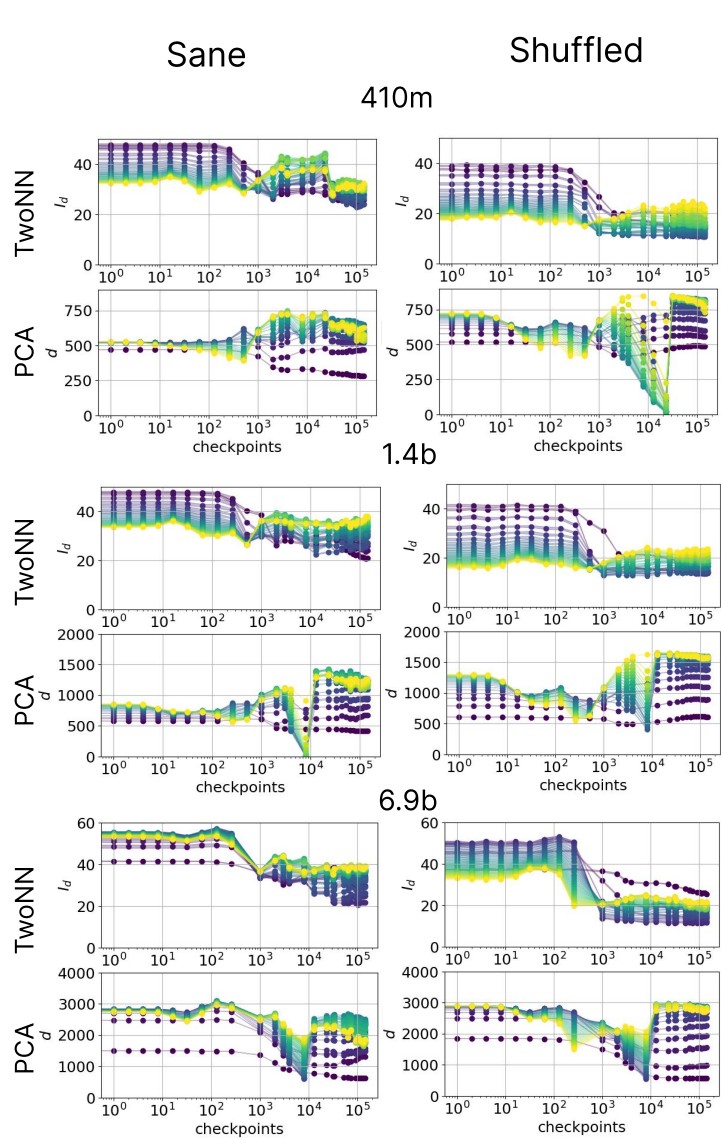

Figure G.5: **Layerwise feature complexity evolution over time, additional results.** Nonlinear $I_d$ (top) and linear $d$ (bottom) over training is shown for coherent (left) and shuffled (right) text, for the 1-coupled setting. Each curve is one layer of the LM (yellow is later, purple is earlier). All settings in [TwoNN, PCA]×[coherent, shuffled] exhibit a phase transition in representational dimensionality at around checkpoint $10^3$, which corresponds to the sharp increase in task performance. In the nonlinear case (top row), the difference between layers' $I_d$ is *low* at the end of training for shuffled text, and *high* for coherent text. This suggests LM learns to perform meaningful and specialized processing over layers. The difference between layers' $d$ (bottom row) at the end of training is, conversely, *high* for shuffled and *lower* for coherent text. This is consistent with our interpretation of $d$ as capturing implied dataset size.

# H ADDITIONAL RESULTS: THE PILE

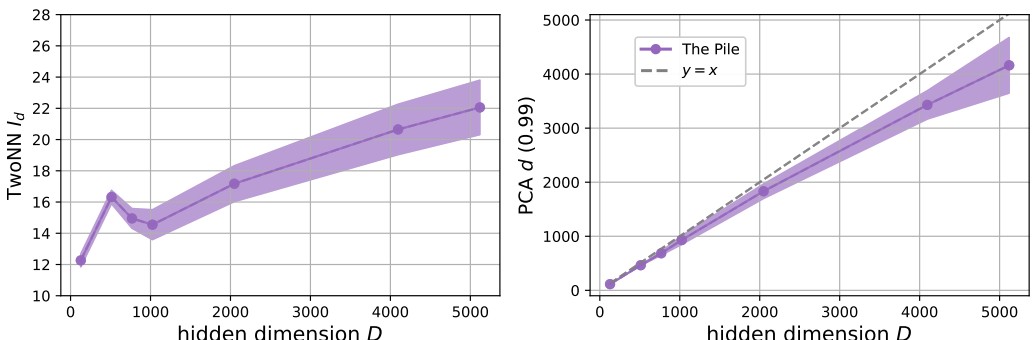

Figure H.1: **Mean dimensionality on the Pile over model size.** Mean nonlinear $I_d$ computed with TwoNN (left) and linear $d$ computed with PCA (right) over layers is shown for increasing LM hidden dimension $D$. TwoNN $I_d$ grows very slowly with extrinsic dimension $D$, while PCA $d$ grows to be nearly one-to-one with $D$. Curves are averaged over 5 random data splits, shown with $\pm 1$ SD.

| | PCA $d$ | | | TwoNN $I_d$ | |
|---|---|---|---|---|---|
| $\alpha$ | **R** | **p-value** | $\alpha$ | **R** | **p-value** |
| 0.8119 | 0.9993 | $2.39 \times 10^{-8}$ | 0.00173 | 0.9537 | $8.64 \times 10^{-4}$ |

Table H.1: **Linear regression of Pythia's average layerwise dimensionality on The Pile to hidden dimension,** $D$. For dimensionality measures (PCA, TwoNN columns), the linear effect size $\alpha$ along with $R$-value and $p$-value are reported. PCA linear dimension shows a statistically significant linear relationship to $D$, with large effect size $\alpha = 0.81$. TwoNN intrinsic dimension also shows a slightly weaker, but still highly significant, linear relationship to $D$. But, the effect size $\alpha$ is near-zero, showing that nonlinear $I_d$ is robust to changes in hidden dimension $D$.

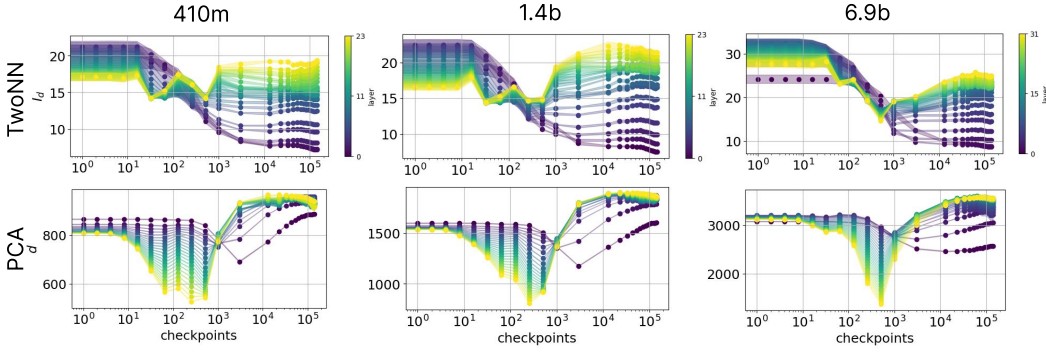

Figure H.2: **ID phase transition in The Pile.** Nonlinear $I_d$ (top) and linear $d$ (bottom) over training is shown for model sizes 410m, 1.4b, and 6.9b (left to right), for The Pile. Each curve is one layer of the LM (yellow is later, purple is earlier). Representations of The Pile exhibit a phase transition in both $I_d$ and $d$ at slightly before checkpoint $10^3$, where $t = 10^3$ corresponds to a dip and redistribution of layerwise dimensionality, and also a sharp increase in task performance in Figure 3.

Table I.1: //++ SEQUENCE LENGTHS// **Spearman correlations between dimensionality and estimated Kolmogorov complexity, varying sequence length**. The Spearman correlation $\rho$ between the `gzip`ped dataset size (KB) and representational dimensionality (rows), averaged over layers, is shown for all tested **Pythia** model sizes (model name omitted for readability). Values marked with a * are significant with a p-value threshold of 0.05. Values marked with † are significant with a p-value threshold of 0.1. Across models, average-layer $I_d$ is not correlated to the estimated Kolmogorov complexity, or formal compositionality, of datasets. Average-layer **linear** $d$ is consistently highly positively correlated to the estimated Kolmogorov complexity. Length $l = 5$ is grayed out as, due to the sequence length being too short, it was not possible to varying the coupling factor $k$; here, the only comparison is between coherent and shuffled ($n = 2$).

| | | sequence length (words) | | | | |
|---|---|---|---|---|---|---|
| | | 5 | 8 | 11 | 15 | 17 |
| 14m | $I_d$ | 1.00 | 0.89 | **0.87**\* | **0.87**\* | -0.10 |
| | $d$ | 1.00 | 0.89 | **0.87**\* | **0.87**\* | **0.81**\* |
| 70m | $I_d$ | 1.00 | 0.40 | 0.43 | 0.26 | -0.10 |
| | $d$ | 1.00 | **1.00**\* | **1.00**\* | **0.98**\* | **0.98**\* |
| 160m | $I_d$ | -1.00 | 0.00 | 0.19 | 0.00 | -0.21 |
| | $d$ | -1.00 | -0.60 | -0.52 | -0.62 | -0.62 |
| 410m | $I_d$ | -1.00 | 0.40 | 0.40 | 0.26 | 0.14 |
| | $d$ | 1.00 | **1.00**\* | **0.98**\* | **1.00**\* | **1.00**\* |
| 1.4b | $I_d$ | -1.00 | 0.40 | 0.40 | 0.43 | 0.14 |
| | $d$ | 1.00 | **1.00**\* | **0.98**\* | **1.00**\* | **1.00**\* |
| 6.9b | $I_d$ | 1.00 | 0.40 | 0.40 | 0.36 | 0.48 |
| | $d$ | 1.00 | **1.00**\* | **0.98**\* | **1.00**\* | **0.98**\* |
| 12b | $I_d$ | 1.00 | 0.40 | 0.40 | 0.43 | 0.00 |
| | $d$ | 1.00 | **1.00**\* | **0.98**\* | **1.00**\* | **1.00**\* |
| Llama-8b | $I_d$ | 1.00 | 0.40 | 0.19 | 0.00 | -0.02 |
| | $d$ | 1.00 | **1.00**\* | **0.98**\* | **1.00**\* | **0.93**\* |
| Mistral-7b | $I_d$ | 1.00 | 0.40 | 0.40 | 0.00 | 0.29 |
| | $d$ | 1.00 | **1.00**\* | **0.98**\* | **1.00**\* | **0.90**\* |

# I ADDITIONAL RESULTS: CORRELATION WITH KOLMOGOROV COMPLEXITY

## I.1 CORRELATION BETWEEN FORMAL COMPLEXITY AND FEATURE COMPLEXITY IS ROBUST TO SEQUENCE LENGTH

//++ NEW// In Table 1 we showed that, for each model, and on a single dataset ($k = 1$, $l = 17$), linear effective $d$ highly correlates to the estimated formal complexity (KC) using `gzip`. Table I.1 shows that this trend is robust to both model family, model size, and sequence length; average layerwise $d$ is almost perfectly monotonic in the formal complexity of the dataset, seen by high Spearman correlation. In contrast, for none of the sequence lengths is average layerwise $I_d$ monotonic in formal complexity, except for the smallest Pythia model (14m).

## I.2 FORMAL COMPLEXITY VS. AVERAGE-LAYER FEATURE COMPLEXITY ACROSS DATASETS

//++ NEW// Figure I.1 shows the global correlation between feature complexity ($I_d$ and $d$) and formal complexity, estimated with `gzip`. While both nonlinear (top row) and linear (bottom row) dimensionality are positively Spearman-correlated to `gzip`, there are clear differences:

1. Linear $d$ increases in the shuffled setting from the coherent setting; nonlinear $I_d$ decreases.

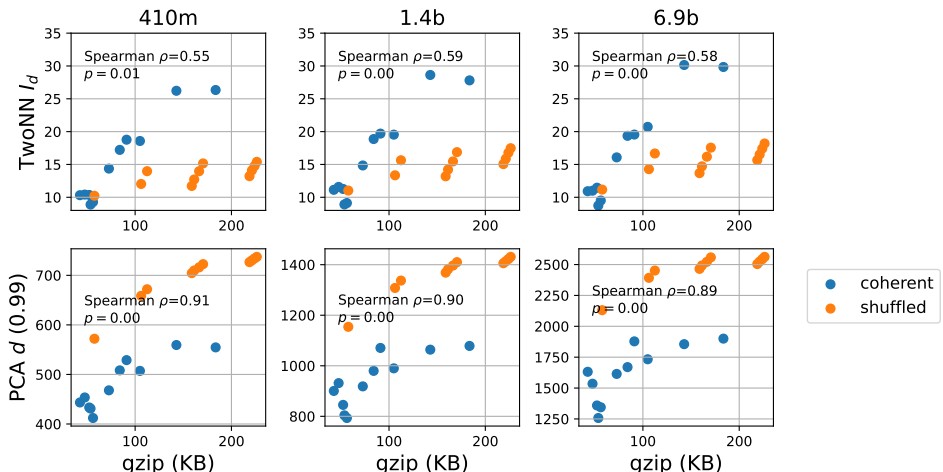

Figure I.1: **Average layerwise dimensionality vs. Estimated Kolmogorov Complexity (gzip)** for Pythia 410m, 1.4b, and 6.9b, aggregated for all grammars. For all models, PCA $d$ highly correlates to `gzip` (estimated KC), with Spearman $\rho \geq 0.9$** for all models. TwoNN $I_d$ correlates more weakly, $\rho \in [0.5, 0.6]$* for all models. Linear $d$ and nonlinear $I_d$ differentially encode shuffled data complexity (orange dots) compared to coherent data complexity (blue dots); where shuffled data display higher $d$ and lower $I_d$. (**) Significant at $\alpha = 0.001$, (*) $\alpha = 0.01$.

    2. Linear $d$ is very highly correlated to the estimated Kolmogorov complexity, $\rho \approx 0.9$ in all cases, while nonlinear $d$ is more weakly correlated, $\rho \in [0.5, 0.6]$.

These observations support the hypothesis that linear effective $d$ encodes formal complexity, while the intrinsic dimension $I_d$ encodes sequence-level semantic complexity.

## I.3   PER-LAYER CORRELATION WITH KOLMOGOROV COMPLEXITY

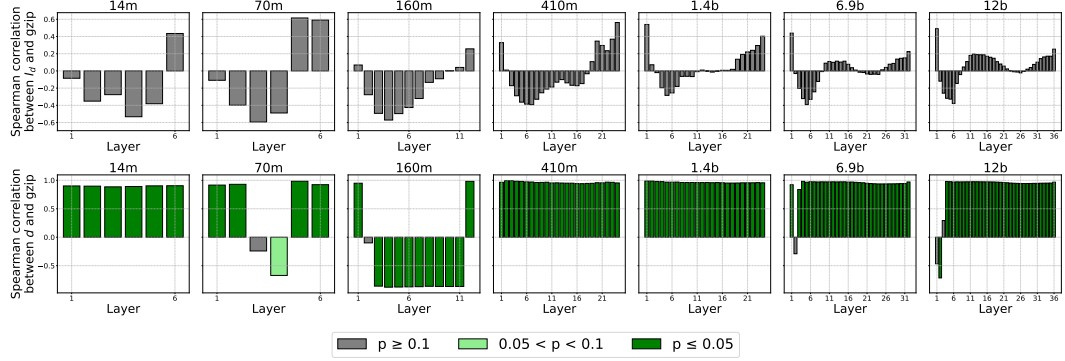

Figure I.2: **Spearman correlations between per-layer dimensionality and estimated Kolmogorov complexity, Pythia models.** The Spearman correlation between the gzipped dataset size (KB) and representational dimensionality per layer, is shown for all tested model sizes for the longest sequence length ($l = 17$). Generally across models, per-layer $I_d$ is not correlated to the estimated Kolmogorov complexity, or formal compositionality, of datasets. Per-layer linear $d$ is consistently highly positively correlated to the estimated Kolmogorov complexity, except one outlier (160m).

**Linear effective $d$ encodes formal complexity robustly across models and datasets**    Figure I.4 shows, for each model, the Spearman correlation between layer dimension and Kolmogorov complexity (`gzip`). Orange boxplots correspond to $d$, and blue boxplots to the $I_d$. Each datapoint in a boxplot reports the correlation for one (model, layer, sequence length) combination; the only factor

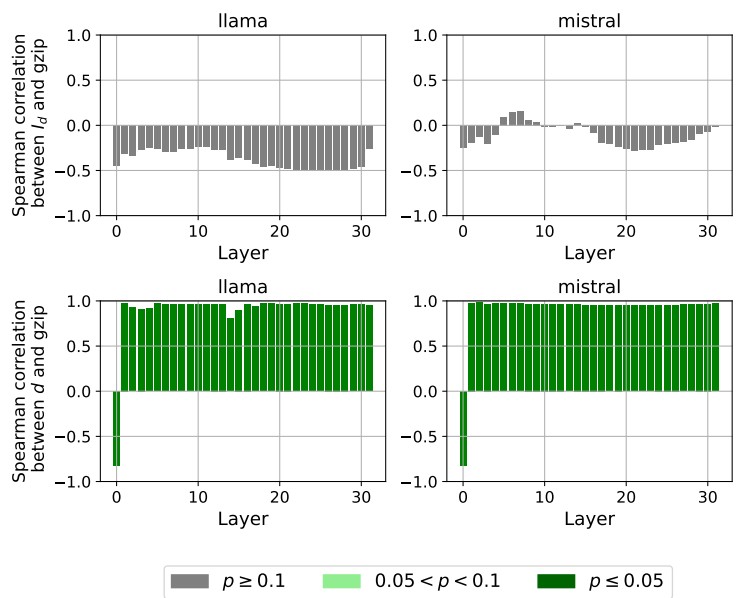

Figure I.3: **//++ NEW//** **Spearman correlations between per-layer dimensionality and estimated Kolmogorov complexity, Llama-3-8B and Mistral-7B.** The Spearman correlation between the gzipped dataset size (KB) and representational dimensionality per layer, is shown for Llama (left) and Mistral (right). Consistently across models, mirroring trends for Pythia, per-layer $I_d$ is not correlated to the estimated Kolmogorov complexity, or formal compositionality, of datasets. Per-layer linear $d$ is consistently highly positively correlated to the estimated Kolmogorov complexity.

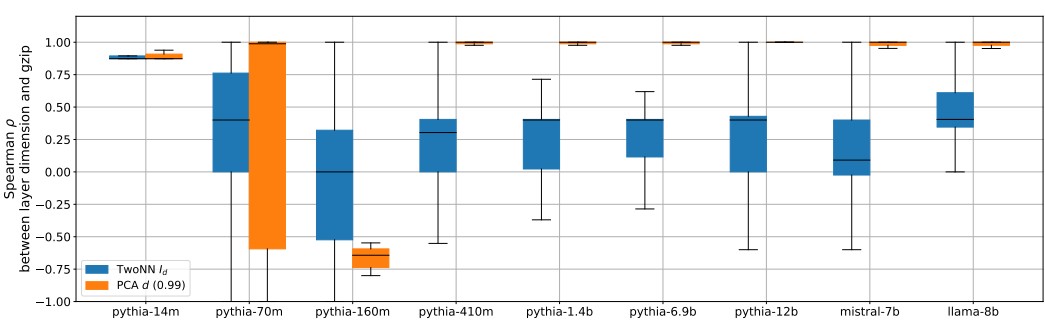

Figure I.4: //++ NEW// **Per-layer Spearman $\rho$ between feature complexity and formal complexity for all models, across all tested datasets.** The layerwise Spearman $\rho$ between formal complexity, measured with `gzip`, and feature complexity, measured with TwoNN $I_d$ (blue) and PCA $d$ (orange), is shown for each model. Each datapoint in each distribution corresponds to one (model, dataset, layer) triple. Generally across models, except for the outlier Pythia-160m, the layerwise correlation between $I_d$ and formal complexity is low, while the correlation to $d$ is high and close to $1.0$ for the vast majority of layers, datasets, and models (orange distributions near $1.0$). This shows that, with high generality across models and datasets, the vast majority of layers encode formal complexity in linear effective $d$, not in the intrinsic dimension $I_d$. Trends are especially robust after a certain model size ($\geq$410m).

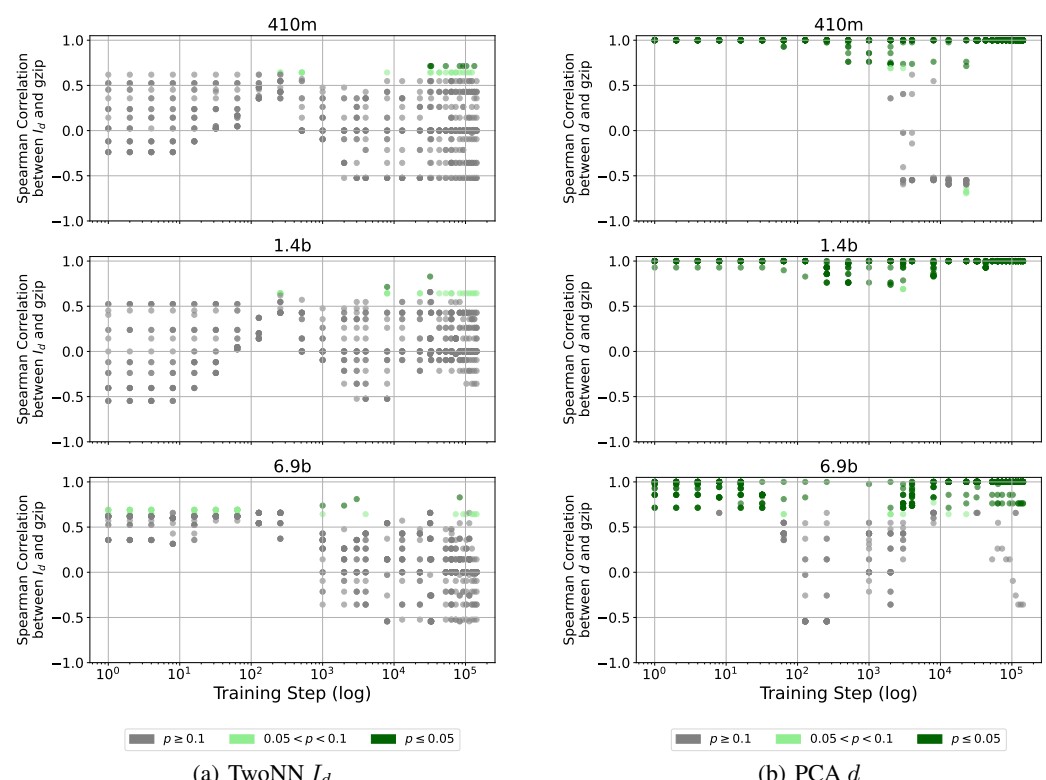

Figure I.5: **Spearman correlation of layerwise dimensionality and Kolmogorov complexity over training.** The Spearman correlations between $I_d$ (left) and `gzip` and $d$ (right) and `gzip` are plotted for three models, 410m, 1.4b, 6.9b (top to bottom) over training time (x axis), where correlation is computed across the controlled corpora. Each vertical set of points denotes the layer distribution of Spearman correlations at a single timestep; each point is one layer's Spearman correlation, colored green if statistically significant and gray otherwise.

of variation in each correlation is the $k$-coupling factor and whether the dataset is shuffled. **With high generality across models and grammars, the linear effective $d$ is monotonic in formal complexity**, seen by the vast majority of layers (orange distributions) close to $\rho = 1.0$ (y-axis). Meanwhile, the $I_d$ does not consistently encode formal complexity, seen by the blue distributions landing about 0.0.

**Outliers** There was one significant outlier, 160m, in our analysis correlating layerwise dimensionality to `gzip` (Kolmogorov complexity), see Figures I.2 and I.4 and Table I.1. While other models consistently demonstrate a positive Spearman correlation between $d$ and `gzip` across layers, 160m (and to a smaller extent, 70m) deviates from this pattern. The reason 160m displays a negative correlation is due to its behavior on shuffled corpora, see the third column in Figure G.1: for intermediate layers, PCA with a variance threshold of 0.99 yields fewer than 50 PCs. We found that this was due to the existence of so-called "rogue dimensions" (Timkey & van Schijndel, 2021; Machina & Mercer, 2024; Rudman et al., 2023), where very few dimensions have outsized norms. Outlier dimensions have been found, via mechanistic interpretability analyses, to serve as a "sink" for uncertainty, and are associated to very frequent tokens in the training data (Puccetti et al., 2022). See Rudman et al. (2023) for exact activation profiles for the last-token embeddings in Pythia 70m and 160m. While increasing the variance threshold to 0.999 reduced the effect of rogue dimensions on PCA dimensionality estimation, we decided to keep the threshold at 0.99 for consistent comparison to other models.

**Coding of formal complexity over training**    The Spearman correlations between layerwise dimensionality ($I_d$ and $d$) and estimated Kolmogorov complexity using `gzip`, over training steps, are shown in Figure I.5 for 410m, 1.4b, and 6.9b. Each dot in the figure is a single layer's correlation to `gzip`; each vertical set of dots is the distribution of correlations over layers, at a single timestep of training. Several observations stand out:

1. PCA encodes formal complexity (seen by earlier dots close to $\rho = 1.0$) as an inductive bias of the model architecture. The high correlation for most layers may be unlearned during intermediate checkpoints of model training, seen by the "dip" in gray dots around steps $10^2 \sim 10^3$, but is regained by the end of training for all model sizes. This indicates that encoding formal complexity at the end of training is a *learned behavior*.

2. TwoNN $I_d$ does not statistically significantly correlate to `gzip` at any point during training, for virtually all layers.

3. For $I_d$, the phase transition noted in Section 4.2 is also present at slightly before $t = 10^3$; this is seen by layerwise correlations in Figure I.5a coalescing to around $\rho = 0.5$, and then redistributing. The layers that best encode formal complexity for TwoNN at the end of trianing correspond to model-initial and model-final layers, see Figure I.2 top.

# J   ADDITIONAL RESULTS: VARYING SEQUENCE LENGTH

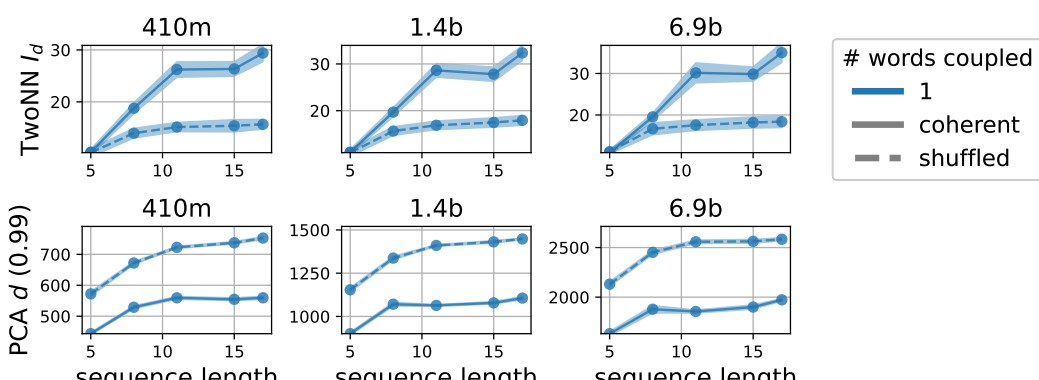

Figure J.1: **Feature complexity increases over sequence length**. The mean $I_d$ and $d$ over layers (y-axis) is shown for increasing sequence lengths $\in \{5, 8, 11, 15, 17\}$ (x-axis) for Pythia models $\in \{410\text{m}, 1.4\text{b}, 6.9\text{b}\}$ (left to right), for the $k = 1$, or the original dataset configuration. Solid curves correspond to coherent, and dashed to shuffled, text. All curves are shown $\pm 1\text{SD}$ over 5 random seeds. Y-axes are scaled to the minimum and maximum for each plot for readability. All curves increase from left to right, evidencing that both nonlinear and linear feature complexity increase with sequence length. Moreover, all curves *saturate*, or plateau, around length=11, indicating this dependence is sublinear.

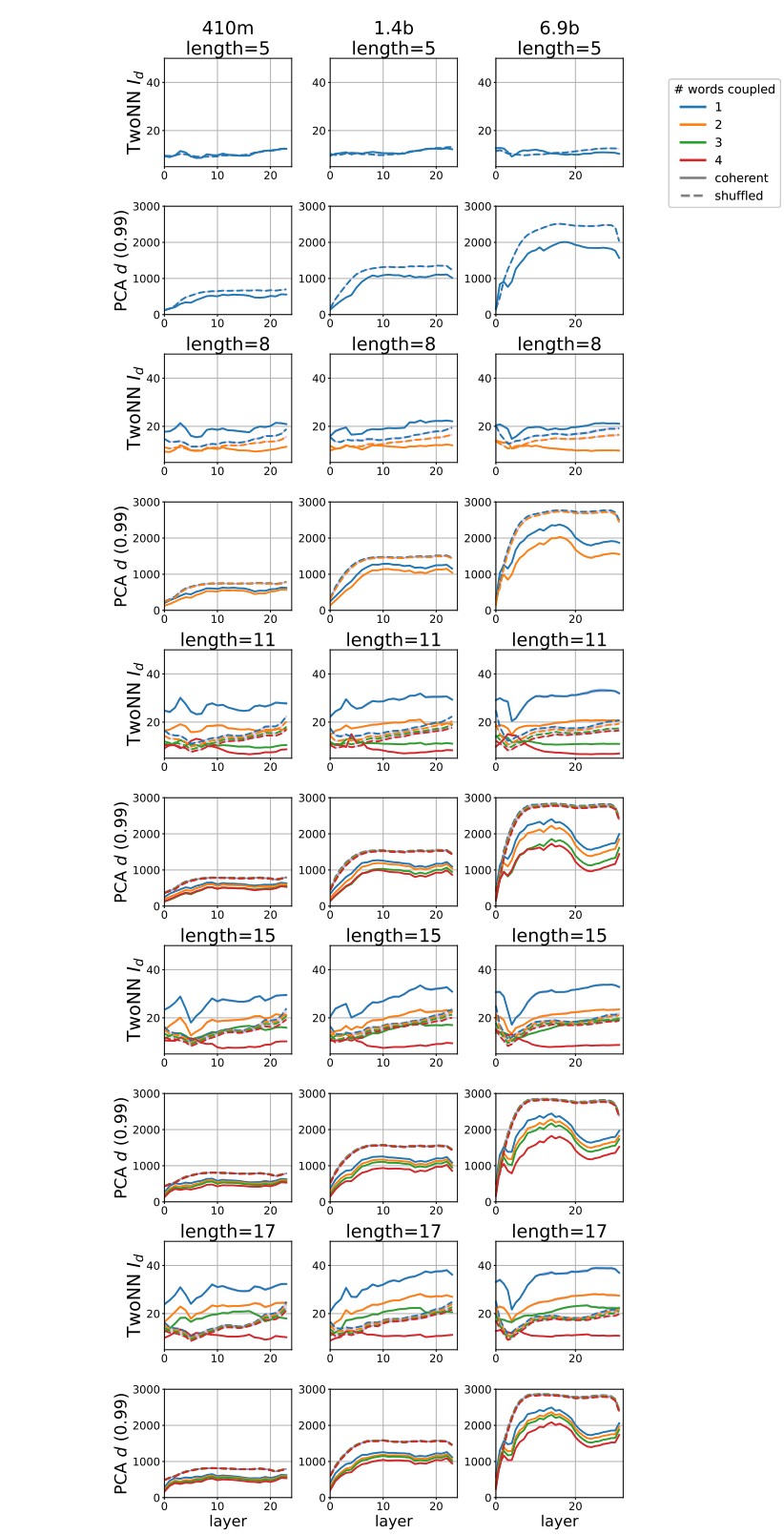

Figure J.2: **Dimensionality over layers, varying sequence length**. Pythia models $\in \{410m, 1.4b, 6.9b\}$ (left to right) and sentence lengths $\in \{5, 8, 11, 15, 17\}$, (top to bottom). In all settings, $I_d$ and $d$ monotonically decrease in $k$; upon shuffling, $I_d$ collapses to a low range while $d$ increases.

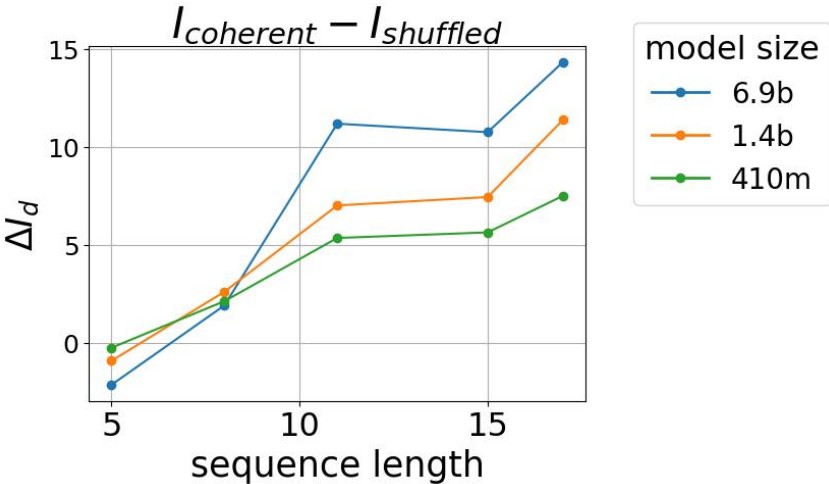

Figure J.3: **Differential coding of semantic complexity increases with sequence length**. The $\Delta(I_d)$ between coherent and shuffled text (y-axis) is shown for Pythia models $\in \{410m, 1.4b, 6.9b\}$ (different curves), as a function of sentence length $\in \{5, 8, 11, 15, 17\}$, (x-axis). For all models, $\Delta(I_d)$ increases as the sequence length increases. For the shortest sequence length $l = 5$, the $\Delta(I_d) \approx 0$, suggesting that at short lengths, (semantic) representational complexity proxies that of a bag of words.

