# OpenReview forum: "GEOMETRIC SIGNATURES OF COMPOSITIONALITY ACROSS A LANGUAGE MODEL’S LIFETIME"
_ICLR.cc/2025/Conference — ICLR 2025 Conference Withdrawn Submission_

### Official Review · Reviewer_U48G · 2024-10-31

**Soundness:** 2
**Presentation:** 2
**Contribution:** 3
**Rating:** 5
**Confidence:** 4

**Summary:**

This paper evaluates the intrinsic dimensionality (linear and nonlinear) of causal language model representations throughout pretraining, for datasets including form composition (combinatorial complexity) and meaning composition (semantic complexity). The models exhibit similar nonlinear intrinsic dimensionalities across model sizes, although linear intrinsic dimensionalities increase for larger models. There is a phase change in nonlinear intrinsic dimensionality around when the models start increasing in performance on several downstream tasks. Linear intrinsic dimensionality is generally larger for shuffled sentences (indexing combinatorial complexity), while nonlinear intrinsic dimensionality is larger for unshuffled sentences (indexing semantic complexity).

**Strengths:**

1. The methods used to measure linear and nonlinear intrinsic dimensionality are well motivated by previous work.
2. The results are interesting, demonstrating the dichotomy between linear intrinsic dimensionality and nonlinear intrinsic dimensionality, where the former indexes combinatorial complexity and the latter indexes semantic complexity.

**Weaknesses:**

Overall, the results are quite interesting, but some of the framing and terminology could be slightly misleading:
1. The paper distinguishes between "form compositionality" and "meaning compositionality". Most existing work on compositionality (https://plato.stanford.edu/entries/compositionality/) defines compositionality as what this paper calls "meaning compositionality" (constituent meanings combine systematically to produce sentence meaning). "Form compositionality" in the paper, measured by the number of unique word combinations, might not be considered "compositionality" by standard definitions. It might be more intuitive to simply refer to "form compositionality" consistently as "combinatorial complexity".
2. The paper focuses on the intrinsic dimensionality in the models for combinatorial complexity (form, e.g. shuffled sentences) and semantic complexity (meaning, in the unshuffled sentences). The paper doesn't seem to focus much on compositionality itself, i.e. how meaning is constructed from form. Thus, the consistent use of the word "compositionality" could be misleading.
3. "But, as sane sequences are grammatical and semantically coherent, it is guaranteed for sane datasets that meaning complexity is monotonic in form complexity. In addition, as shuffling removes sequence-level semantics, meaning complexity is guaranteed to be lower on shuffled compared to sane text, by definition" (p. 5). These sentences implicitly assume a definition of "meaning complexity". E.g. increasing form complexity does not necessarily guarantee an increase in meaning complexity (e.g. "It is possible that rain will occur today" vs. "It might rain today"), and defining shuffled sentences to be "low meaning complexity" is up to the definition of meaning complexity. These assumptions still seem safe for the conclusions in the paper, but the assumption of this specific definition of semantic complexity should be noted.
4. Minor point: "Our stimulus dataset consists of grammatical nonce sentences from the grammar illustrated in Figure 1" (p. 3). In linguistics, nonce sentences usually refer to grammatical but semantically incoherent sentences, rather than semantically coherent sentences as in this dataset. Removing the word "nonce" might be more clear.
5. Minor note on terminology: calling the unshuffled sentences "coherent sentences" or "original sentences" would be more in line with existing work than "sane sentences" (e.g. https://arxiv.org/pdf/1803.11138).

**Questions:**

1. Rather than an effect of lower semantic complexity, could the low nonlinear intrinsic dimensionality of shuffled sentences simply be because they're out of distribution? E.g. the model would likely allocate more "representation space" to in-distribution data (i.e. the unshuffled/coherent sentences). Then, a parsimonious explanation of the results could just be that lower k for unshuffled sentences increases nonlinear intrinsic dimensionality due to broader in-distribution semantic diversity, but the effect is not seen for shuffled sentences because those sentences are out-of-distribution (i.e. not allocated space in the model anyways).

---

> ### Author Response · Authors · 2024-11-14
> **Response to reviewer**
>
> Thanks so much for your comments, especially recommendations on the framing. We'll respond to your suggestions below in more detail:
>
> __It might be more intuitive to simply refer to "form compositionality" consistently as "combinatorial complexity". The paper focuses on the intrinsic dimensionality in the models for combinatorial complexity (form, e.g. shuffled sentences) and semantic complexity (meaning, in the unshuffled sentences). The paper doesn't seem to focus much on compositionality itself, i.e. how meaning is constructed from form. Thus, the consistent use of the word "compositionality" could be misleading.__
>
> Thanks for the suggestion. The combinatorial complexity of forms is indeed not the type of compositionality put forth by Frege and in the Szabo chapter. Instead, it refers to the extent to which a language realizes its combinatorial possibilities at the system level, which does have precedence in the literature; see for instance the definition for “system-level compositionality” in “Language Use is Only Sparsely Compositional”, ([Sathe, Federenko and Zaslavsky, 2024](https://escholarship.org/uc/item/0qd3662b)) as well as in Section 3.1 of "A Complexity Theory-Based Definition of Compositionality", ([Elmoznino et al. 2024](https://arxiv.org/abs/2410.14817)).
>
> Still, we think it’d be prudent to change “form compositionality” to “combinatorial complexity” as suggested, and “meaning compositionality”, which is a binary variable (presence/absence of phrase-level semantic composition), to “semantic complexity”.
>
> We propose to change the framing as follows:
> 1. Clearly distinguish between phrase-level semantic composition (Frege, Szabo) and system-level compositionality (Sathe et al., 2024) around line 084. Name these terms respectively semantic complexity and combinatorial complexity.
>
> 2. Make the appropriate changes to the text throughout.
>
> As we don’t want terminology to distract from the contribution of our paper, we’d greatly appreciate your feedback on the current proposal (or if it suffices to keep the current terms while better defining them and motivating them in the literature).

---

> > ### Author Response · Authors · 2024-11-14
> >
> > __…These assumptions still seem safe for the conclusions in the paper, but the assumption of this specific definition of semantic complexity should be noted.__
> >
> > Thanks for this comment. We agree that our conclusions hold in the absence of figurative language / paraphrases as you pointed out. The grammar was constructed to preclude this possibility, and we will point this out explicitly in “Measuring formal and semantic compositionality” in Section 3.2.1.
> >
> > __In linguistics, nonce sentences usually refer to grammatical but semantically incoherent sentences__
> >
> > Thanks for the pointer! We have removed it.

---

> > > ### Author Response · Authors · 2024-11-14
> > >
> > > __Then, a parsimonious explanation of the results could just be that lower k for unshuffled sentences increases nonlinear intrinsic dimensionality due to broader in-distribution semantic diversity__
> > >
> > > Really interesting hypothesis. According to your suggestion we looked at the ID of the controlled grammar inputs, which is more OOD compared to The Pile. The ID of the inputs is actually higher for k=1, which suggests that the lower-ID effect for shuffled sequences isn't simply due to it being out-of-distribution. In future work, it would be interesting to explicitly relate the degree of distribution shift to geometric properties of representations.

---

> ### Comment · Reviewer_U48G · 2024-11-23
>
> Thank you for the responses! I've decided to keep my score unchanged, but I appreciate the work put into the additional experiments. I still feel that the paper does not tie the results to many questions specifically around compositionality, how meaning representations are constructed from form.

---

### Official Review · Reviewer_QP9Q · 2024-10-31

**Soundness:** 2
**Presentation:** 2
**Contribution:** 2
**Rating:** 5
**Confidence:** 5

**Summary:**

This paper investigates the relationship between compositionality and the geometric complexity of language model (LM) representations as a function of the data. Using a controlled dataset with varying levels of compositionality and the Pythia models, the authors explore how intrinsic dimensionality and linear effective dimensionality  change with input compositionality.

**Strengths:**

- The premise is novel and investigates the impact of the degree of compositionality in linguistic input to the intrinsic dimensionality of its representation manifold in LMs.

- Comprehensive analysis of dimensionality measures: The study examines both linear (d) and non-linear (Id) dimensionality measures and how they change with input.

- The study employs a well-defined experimental setup with a custom dataset designed to control compositionality and investigates multiple Pythia model sizes.

**Weaknesses:**

- The discussion of the core concept of compositionality, on which the paper is based, is quite shallow and scattered. Both the introduction and background sections talk about a bottom-up concatenative notion of compositionality (Frege, Chomsky etc) which has been considered unsuitable for discussing compositionality for connectionist architectures (Smolensky 1987, Van Gelder 1990, Chalmers 1993). The investigation of compositionality would require proper characterization of the concept itself.

- The distinction between form vs meaning compositionality needs to be motivated better. Compositionality is about discovering the underlying structure of data (and bottom-up concatenation is just a mere product of this process) so the distinction between compositionality of form and meaning doesn’t make sense in this context since the former is not a representation of compositionality in any sense but rather some kind of a type-token ratio/word statistic tracker.

- On line 107, it mentions “compositionality of inputs” but this phrase is somewhat vague. Different kinds of syntactic and semantic characteristics of inputs are manipulated in the input, but how do these features relate to compositionality?

- For form compositionality, Kolmogorov complexity is used, but why? The decision needs to be justified.

- In lines 227-229, the claim seems like a leap. If we cannot define or quantify semantic complexity (line 226), how do we ascertain a link to ID in models just by a comparative measure? Meaning complexity can and should be quantified to make claims based on what aspects of model representations track form vs. meaning complexity.

- The use of the transformer’s residual stream for analysis is common but it is also one of the most non-privileged in terms of basis according to a lot of mechanistic interpretability literature (Elhage et al 2021 etc). Other representations of different components (Attention, MLP) could be used here instead.

**Questions:**

- Could you provide a more thorough characterization of compositionality as it pertains to connectionist models? How does your understanding align with or diverge from foundational perspectives (e.g., Smolensky, Van Gelder, Chalmers) on compositionality in connectionist architectures? How does the definition of compositionality you use set the foundation for your investigation and analyses throughout the paper?

- Could you clarify and justify the distinction you make between compositionality of form and meaning? What theoretical or empirical motivation supports this separation? Could you explain why tracking form (e.g., type-token ratio) should be considered a component of compositionality rather than a separate linguistic metric?

- Why did you choose Kolmogorov complexity as a metric for form compositionality? What specific advantages does it offer for assessing compositionality, and how does it relate to the core concept of structure discovery?

- Could you elaborate on the reasoning behind linking semantic complexity to Identification (ID) in models, particularly in the absence of a clear semantic complexity metric? Is there a way to quantitatively define or approximate semantic complexity to strengthen claims about its relationship with model representation?

- Why did you prioritize the transformer’s residual stream for your analysis? How does it align with or diverge from the representational basis recommended in mechanistic interpretability literature? Have you considered incorporating other components, such as Attention or MLP layers, into your analysis?

---

> ### Author Response · Authors · 2024-11-14
> **Response to reviewer**
>
> Thanks for your valuable feedback! We’ve responded in detail to each of your comments below.

---

> ### Author Response · Authors · 2024-11-14
> **Response to Comment 1 (link to connectionist models)**
>
> __Could you provide a more thorough characterization of compositionality as it pertains to connectionist models? How does your understanding align with or diverge from foundational perspectives (e.g., Smolensky, Van Gelder, Chalmers) on compositionality in connectionist architectures? How does the definition of compositionality you use set the foundation for your investigation and analyses throughout the paper?__
>
> Thanks for this question. The specific perspectives of Smolensky etc on meaning composition in connectionism prescribe how meaning composition and binding occur in distributed representations.
>
> In brief, our work is somewhat unrelated to composition in the connectionist sense; we do not investigate the composition function mapping _representations_ of parts to the meaning _representation_ of the whole. Our question is more simple: does the geometric complexity of representations correlate to the (formal, semantic) compositional complexity of inputs? As a result, we only consider compositionality on inputs, with representational dimension as a dependent variable; the link to connectionism would instead have to consider compositional structure of representations.

---

> ### Author Response · Authors · 2024-11-14
> **Response to Comment 2 (Distinction between compositionality of form and meaning)**
>
> __Could you clarify and justify the distinction you make between compositionality of form and meaning? What theoretical or empirical motivation supports this separation? Could you explain why tracking form (e.g., type-token ratio) should be considered a component of compositionality rather than a separate linguistic metric?__
>
> We will be careful to clarify these two notions of compositionality in the manuscript.
>
> Our “meaning compositionality” is the one of Frege, Partee, etc. The familiar definition by Szabo is “the meaning of a complex expression is fully determined by its structure and the meanings of its constituents” or the “bottom-up concatenative notion of compositionality” you refer to. This has been referred to as semantic compositionality in the literature [(Baroni, 2019)](https://royalsocietypublishing.org/doi/full/10.1098/rstb.2019.0307), [(Pelletier, 1994)](https://link.springer.com/article/10.1007/BF00763644) among others.
>
> Our “formal compositionality” is the system-level compositionality described in “Language Use is Only Sparsely Compositional”, [(Sathe, Federenko and Zaslavsky, 2024)](https://escholarship.org/uc/item/0qd3662b) as well as [(Elmoznino et al., 2024)](https://arxiv.org/abs/2410.14817). It refers to the extent to which a language realizes its combinatorial possibilities, and is what we control with $k$. The type-token ratio of $k$-grams, in this case, is related. A lower type-token ratio indicates the dataset is “less compositional” according to the system-level definition. We’ll add this interpretation to the manuscript.
>
> We recognize that the more familiar definition of compositionality to readers is semantic compositionality. To avoid confusion, we’d appreciate your feedback on the following changes to the manuscript:
>
> - Either we change the terms “formal compositionality” to “combinatorial complexity”, and “semantic compositionality” -> “semantic complexity” to avoid terminology confusion, adding the above citations and motivation to the Background, or
> - Keep the terms as-is, adding the above citations and discussions to Background, around line 084.

---

> ### Author Response · Authors · 2024-11-14
> **Response to Comment 3 (Why Kolmogorov Complexity)**
>
> __Why did you choose Kolmogorov complexity as a metric for form compositionality? What specific advantages does it offer for assessing compositionality, and how does it relate to the core concept of structure discovery?__
>
> Kolmogorov complexity (KC), coming from algorithmic information theory, is the length of the shortest program (in bits) needed to produce the dataset. If a dataset is structured, such as in our case, then the program length will be shorter than for a dataset that appears random (without structure). In other words, all else being equal, a less combinatorially complex (in our case “formally compositional”) dataset is more compressible.
>
> Using KC to relate linguistic structure to compression is not that new. Along these lines, recent work has proposed Kolmogorov complexity as a representation-agnostic measure of compositionality (Elmoznino et al., 2024); others have related information-theoretic compression more broadly to language learning in machines (e.g. Deletang et al., 2023) and learning and comprehension in humans (Mollica and Piantadosi, 2019; Levy, 2008), as well as system-level compositionality of languages (Sathe et al., 2024). The throughline in all of these works is that the compositional structure of language permits compression, making KC highly suited to quantify formal compositionality.
>
> We will add the motivation and relevant citations around line 220 in "Measuring formal and semantic complexity".

---

> ### Author Response · Authors · 2024-11-14
> **Response to Comment 4 (Why link semantic complexity to ID)**
>
> __Could you elaborate on the reasoning behind linking semantic complexity to ID in models, particularly in the absence of a clear semantic complexity metric? Is there a way to quantitatively define or approximate semantic complexity to strengthen claims about its relationship with model representation?__
>
> Quantifying semantic complexity is highly nontrivial, as for general text datasets, it cannot be directly inferred from the complexity of the strings, see e.g. the provided example [cat, puma, lion] in line 224. In particular, semantic complexity must be quantified on some meaning representation of the input strings, which would require, e.g., embedding the sentence using a large language model. As we are comparing semantic complexity to dimensionality computed on LM embeddings, this would unfortunately be circular.
>
> Instead, we constructed our grammar such that sentences’ meanings are literal compositions of the meanings of their parts. This guarantees that a dataset with a larger variety of word combinations (smaller $k$) has a larger variety of meanings. That is, semantic complexity, where we consider semantics to be at the sentence-level, monotonically decreases in $k$ for normal text; this is enough as we only care about the relative complexity between datasets. On the other hand, sentence-level semantic complexity should collapse when shuffling sequences, as by construction, destroying syntax removes the possibility of composing phrase-level meaning.
>
> We mention this reasoning in line 228, but will also add the above explicitly in the paper.

---

> ### Author Response · Authors · 2024-11-14
> **Response to Comment 5 (Why use the transformer's residual stream)**
>
> __Why did you prioritize the transformer’s residual stream for your analysis? How does it align with or diverge from the representational basis recommended in mechanistic interpretability literature? Have you considered incorporating other components, such as Attention or MLP layers, into your analysis?__
>
> Looking at the contribution of attention to ID and MLP could be an exciting and novel direction for follow-up work. But, due to several reasons detailed below (one of them being compute), we did not prioritize it in this project.
>
> Analyzing the residual stream (Elhage, 2021) is a very common paradigm in the interpretability literature. It has a lot of nice interpretations, such as an iterative refinement of vocabulary distributions (Geva et al, 2022; Belrose et al, 2023; Dar et al, 2022) until the next-token prediction; or a communication channel onto which individual layers write (Elhage et al., 2021; Merullo et al., 2024). We prioritize the residual stream over more fine-grained components in this work for several reasons. In particular, mechanistic interpretability, which would analyze individual neurons and circuits, up to attention heads and MLP layers, is not our goal. As we’re looking at dataset-level properties, we are interested in representations that are coarse-grained: this makes the residual stream an ideal candidate compared to, e.g.,  individual attention heads. We also care about comparability to the existing literature on ID estimation in LMs. The vast majority of existing works (Valeriani et al. 2023; Cheng et al. 2023; Cai et al. 2021; Chen et al. 2020; Antonello et al., 2024; Doimo et al., 2024; Tulchinskii et al, 2023; Yin et al, 2024) compute geometric measures such as ID at the level of the residual stream, and specifically on the last token representation. As such, our work is in line with existing literature.
>
> We will motivate this explicitly in the Methods section.

---

> ### Author Response · Authors · 2024-11-23
>
> Thank you for your thoughtful comments again.
> As the end of the discussion period is approaching, we are hoping to hear about your opinion and thoughts on our response.

---

> > ### Comment · Reviewer_QP9Q · 2024-11-23
> > **Official Comment**
> >
> > I have gone through the responses. For the distinction of compositionality between form and meaning I would suggest going with complexity rather than "compositionality" as suggested.  However, overall I would like to keep the score as it is given the overall premise of the paper.

---

### Official Review · Reviewer_vTcJ · 2024-11-03

**Soundness:** 3
**Presentation:** 3
**Contribution:** 2
**Rating:** 6
**Confidence:** 4

**Summary:**

The goal of the paper is to relate compositionality to high-level dimensionality heuristics. For that the authors create a dataset with sentences syntactically identical sentences using a simple grammar. They extend such dataset by modifying its combinatorial complexity by correlating different numbers of words within the sentences. The authors also create a shuffled version of the dataset also with different combinatorial complexity levels which they use as control throughout the paper.

The paper finds that linear dimensionality is a proxy for form whereas nonlinear dimensionality encodes meaning. This is supported by a large set of experiments on models from the Pythia family, evaluated on both the controlled dataset and The Pile.

**Strengths:**

1. **Clarity and Organization**: The motivations, research questions, and methodologies are presented clearly
2. **Comprehensive Literature Review**: The literature is thoroughly reviewed, framing the research well in the context of existing work on compositionality and intrinsic dimensionality.
3. **Extensive and Organized Results**: The paper includes a large set of results, which are mostly well presented.

**Weaknesses:**

1. **Model Choice**:
   - The paper uses Pythia, which is not state-of-the-art (SOTA), though it does have the advantage of available checkpoints.
   - It would strengthen the paper to include SOTA models, such as those from the Llama or Mistral families, to see if the findings generalize to the most current models.
   - **Suggestions**: Run experiments on a final checkpoint for the SOTA models. Observing differences across layers in shuffled versus unshuffled data, along with gzip correlation results.


2. **Dataset Limitations**:
   - The controlled dataset is restricted to a single syntactic structure, which may limit the generality of the findings.
   - To explore the effects on a more diverse linguistic structure, it would be useful to introduce additional syntactic forms, where compositionality varies by grammar rather than by word correlations.
   - **Suggestions**: Extend the dataset to include varied syntactic structures that capture additional linguistic features, such as syntactic depth, constituent length, and sentence length, and observe how linear and nonlinear dimensionality metrics respond to these variations.


3. **Literature Integration**:
   - The paper does not adequately address previous work on linear probing and syntactic encoding in linear subspaces (such as Hewitt and Manning, 2019).
   - **Suggestions**: Discuss the relevance of linear probing work, which demonstrates that syntax is encoded in a linear subspace, to the findings here.

**Questions:**

1. **Interpretation of Dimensionality Collapse (Figure G.4)**:
   - There is a collapse for \(d\) around checkpoint \(10^4\) in models with 1.4b and 6.9b parameters, but not for “The Pile.”
   - **Question**: What does this collapse represent? Does it indicate that the phase transition for encoding meaning occurs earlier than for encoding structure?

2. **Choice of Aggregation Over the Last Token**:
   - The paper states, “We aggregate over the sequence by taking the last token representation, as, due to causal attention, it is the only to attend to the entire context.”
   - **Question**: Why did you choose this aggregation method instead of tracking dimensionality metrics (Id and d) incrementally for all tokens within the sentence?

---

> ### Author Response · Authors · 2024-11-15
> **Response to Reviewer**
>
> Thanks for your valuable feedback! We respond to each comment in detail below.

---

> > ### Author Response · Authors · 2024-11-15
> > **Llama and Mistral replication study**
> >
> > __It would strengthen the paper to include SOTA models, such as those from the Llama or Mistral families, to see if the findings generalize to the most current models. Run experiments on a final checkpoint for the SOTA models. Observing differences across layers in shuffled versus unshuffled data, along with gzip correlation results.__
> >
> > Thank you for this suggestion! We have already replicated our findings on Llama-3-8B and Mistral-7B; results were robust. We find that
> > 1.  $I_d$ and $d$ negatively depend on $k$ as for Pythia;
> >
> > 2. $d \sim gzip$ while $I_d$ is not correlated to gzip (formal complexity);
> >
> > 3. Shuffling sequences, removing sequence-level semantics, collapses $I_d$ to a low range while increasing $d$.
> >
> > We posted a new version of the pdf where the analogue to Fig 4 for Mistral and Llama is found in Fig G.2, and the new Kolmogorov complexity correlations per layer in Fig. I.2.

---

> > > ### Author Response · Authors · 2024-11-15
> > > **Extension to other syntactic structures**
> > >
> > > __Extend the dataset to include varied syntactic structures that capture additional linguistic features, such as syntactic depth, constituent length, and sentence length, and observe how linear and nonlinear dimensionality metrics respond to these variations.__
> > >
> > > Very interesting suggestion. Thus far, as recommended, we are running experiments on grammars of varying sentence length that are similar to the current one. On these new grammars, we are similarly controlling the input complexity via coupling $k$-grams, which tunes the datasets’ combinatorial complexity that we’re interested in.Stay tuned for these results; we’ll ping you when we upload a new version of the manuscript.
> > >
> > > On the other hand, creating dataset variations with different levels of recursive embedding and constituent length, while keeping all else equal, seems highly nontrivial, and would be an interesting challenge for future work. In particular, it’s not hard to imagine that dataset combinatorial complexity could impact the ID of representations. But, we hypothesize that the geometry of recursive center embedding, for example, may be represented elsewhere than in the last token (e.g., it might be in the “border tokens” of each recursion), and may require different geometric tools. We’re happy to hear further thoughts on this.

---

> > > > ### Author Response · Authors · 2024-11-15
> > > >
> > > > __Discuss the relevance of linear probing work, which demonstrates that syntax is encoded in a linear subspace, to the findings here.__
> > > >
> > > > Thanks for the suggestion! We will include Hewitt and Manning (2019) around line 100 where we discuss encoding of linguistic structure in representations.

---

> > > > > ### Author Response · Authors · 2024-11-15
> > > > >
> > > > > __Why did you choose this aggregation method instead of tracking dimensionality metrics (Id and d) incrementally for all tokens within the sentence?__
> > > > >
> > > > > We did not track the geometry of intermediate tokens for a couple reasons. First, because tokenization results in sequences of slightly different lengths, we cannot guarantee that each token position has the same number of datapoints, making the token manifolds incomparable. Second, because tokens are often subwords, the meaning of the dimensionality at a certain token position, e.g., the 4th token, is unclear. For instance, the 4th token could mark the end of the following sequences: "I_ went_ to_ a_", "I_ went_ to_ the_", and "I_ went_ to_ Mar_", where the continuation of the third sequence is "I went to Marie's house"; in the third case, the 4th token position corresponds to a linguistically vacuous unit, i.e. not a word.
> > > > >
> > > > > Still, we were very intrigued by your comment. We’re running an experiment on Pythia models 410m, 1.4b, and 6.9b (the main paper models; last training step) to see how feature complexity changes along tokens _corresponding to the ends of words_ in the sentence. That is, to avoid the problems we mentioned above, we need to do an extra step that matches token indices to word boundaries. This may take a while, and we’ll update when we have the results.
> > > > >
> > > > > Otherwise, we prioritized the last token representation for several reasons. The last token representation in the residual stream is the default sentence embedding in the interpretability literature. The reason the last token (and [CLS] in BERT) is the go-to sentence embedding is because it is the only to attend to the whole context, and is the one eventually used for next-token prediction– see (Geva et al., 2022; Dar et al., 2022; Elhage et al., 2021).
> > > > >
> > > > > It is also the default in the existing literature on ID estimation in LMs. The vast majority of existing works (Cheng et al. 2023; Cai et al. 2021; Chen et al. 2020; Antonello et al., 2024; Doimo et al., 2024; Tulchinskii et al, 2023; Yin et al, 2024) compute geometric measures such as ID at the level of the residual stream, and specifically on the last token representation. Your suggested experiment would constitute the first results on intermediate tokens, as far as we are aware.

---

> ### Author Response · Authors · 2024-11-18
>
> ___What does this collapse represent? Does it indicate that the phase transition for encoding meaning occurs earlier than for encoding structure?___
>
> Signature of encoding structure was given by a relative difference between sane/shuffled, not the absolute value of either individually. Around $10^4$ the profiles across the layers for both sane/shuffled settings looks the same. Therefore the formal complexity between normal and shuffled text is not differentiated . The differential coding is what signifies learning of formal complexity. If we look around $10^2$-$10^3$ there is indeed a first dip, where the differential encoding between coherent and shuffled text can first be seen.

---

> ### Author Response · Authors · 2024-11-23
>
> Thank you for your comment and feedback again!
>
> As the end of the discussion period is approaching, we would like to hear your opinion on our response and additional results which are coherent with our previous results and additional insight on emergence of complexity as a function of sequence length, as summarized in a global response on the top:
> https://openreview.net/forum?id=q5lJxCXjiY&noteId=KZEbFhTUaV

---

> > ### Comment · Reviewer_vTcJ · 2024-11-24
> >
> > Thank you for your clarification comments and the additional experiments run. I have decided to keep my score unchanged.
> >
> > I share the concern about the term 'compositionality', instead the paper focuses on 'combinatorial complexity' which changes its scope and the implications. In my opinion, addressing the current hypothesis (on compositionality) requires some technical work (on tokenization and aggregation) and a carefully designed set of synthetic datasets controlling for different sentence lengths and linguistic features.

---

### Official Review · Reviewer_gP62 · 2024-11-05

**Soundness:** 1
**Presentation:** 2
**Contribution:** 1
**Rating:** 3
**Confidence:** 3

**Summary:**

The authors empirically explore compositional language model representations, and how these representations change during training by using stored checkpoints from the Pythia models; outcomes are measured using three different model sizes.

The authors rely on two datasets to measure this. The first is a completely synthetic dataset with unspecified distributional properties from an artificially limited grammar of fixed length. The second consists of randomly selected passages from the Pile, selected without consideration of any reasonable boundaries, and again of a fixed length.

Using the representation of the last token as a proxy for the whole token sequence, the authors try to measure the dimensionality of these datasets, both linearly using PCA and in a non-linear fashion using TwoNN to estimate intrinsic dimension.

The authors then compare model dimension vs empirical data dimension, intrinsic dimension during training, and dimension by layer, drawing some conclusions from these empirical experiments.

**Strengths:**

The exploration of changing dimension across checkpoints was interesting; that felt original.

Some explanation of dimension was relatively clear.

Figure 3 – the change in iD is interesting over epochs; there are phase transitions at points that seem to occur at similar points in training.

**Weaknesses:**

Ostensibly the paper explores compositionality, but the definition of compositionality and its experimental setup were unusual and not well connected to linguistic notions of compositionality. Given datasets that feel unrepresentative of real language distributions, any conclusions drawn from this dataset are unlikely to apply to any real NL data. Furthermore, it was not clear to me how the experiments attempted to measure compositionality.

I had quite a few clarity issues about how the data was constructed, and I felt the authors leapt to conclusions about from relatively scarce data.

In detail:
* The first dataset is a very limited Controlled Grammar – much weaker than even a probabilistic context free grammar. As such, it feels like a very limited exploration into actual linguistic phenomena. I would have expected that a study of compositionality would relate the representation of parts (e.g. words or phrases) to larger constructions (e.g. sentences or paragraphs).
* The authors describe briefly “composition of forms” vs “composition of meaning” – I found these notions unclear, not well connected to any linguistic or ML definition of compositionality. This should either be explained more fully or a citation should be provided.
* The authors say that k contiguous words are coupled during sampling, but they do not describe the sampling distribution.
* Also, I’m not sure how this is supposed to measure compositionality. I could imagine an experiment to evaluate whether changes in nationality or in job led to systematic and predictable differences in the constructed representation, but that was not present.
* From a linguistic standpoint, sampling 16 contiguous tokens is strange. The 16 contiguous tokens might span sentences, paragraphs, etc. Hence even the real data felt unclean and not representative.
* I was concerned about using the last token as a choice to represent the entire sequence. Although it is true that is the only token that can attend to all positions, there is nothing in the training objective that encourages it to represent the complete sequence. Subsequent tokens can attend to any position.
* Figure 2 – it’s odd to me that shuffled distributions have lower intrinsic dimensionality, even in the unigram case. I would have expected the dimensionality would be higher in the case of shuffling, as the data has fewer constraints.

**Questions:**

* What is the sampling distribution and process for generating data? Is an n-gram language model used? Or a neural model with a given window?
* Why not sample utterances from The Pile that are whole sentences instead?
* Why use the last token as the representation? Did the authors evaluate how well that reflects the earlier tokens in the string? Could, for instance, the original string be reconstructed with reasonable likelihood given this representation?
* Figure 2 – is this mostly telling us something about the dimension from which the data was sampled? What happens to these numbers if the number of categories is changed?

---

> ### Author Response · Authors · 2024-11-15
> **Response to Reviewer**
>
> Thanks so much for your careful review of our paper! You raised some great questions and we hope to have resolved them below.

---

> > ### Author Response · Authors · 2024-11-15
> > **Contextualizing compositionality definitions**
> >
> > __the definition of compositionality and its experimental setup were unusual and not well connected to linguistic notions of compositionality… “composition of forms” vs “composition of meaning” – I found these notions unclear, not well connected to any linguistic or ML definition of compositionality. This should either be explained more fully or a citation should be provided.__
> >
> >
> > We apologize for the confusion, and will be careful to better contextualize formal vs. meaning compositionality in the manuscript. Both definitions indeed have precedence in the linguistics literature, where our “meaning compositionality” is a property of language (as noted by us and Reviewer 6) and “formal compositionality” is related to language use.
> >
> >
> > Our “meaning compositionality” is the one of Frege, Partee, etc. The familiar definition by Szabo is “the meaning of a complex expression is fully determined by its structure and the meanings of its constituents”, which we essentially restate in l084 and l198. This has been referred to as semantic compositionality in the literature [(Baroni, 2019)](https://royalsocietypublishing.org/doi/full/10.1098/rstb.2019.0307), [(Pelletier, 1994)](https://link.springer.com/article/10.1007/BF00763644) among others. We will cite the relevant sources and format it in a “Definition” environment.
> >
> >
> > Our “formal compositionality” is the system-level compositionality described in “Language Use is Only Sparsely Compositional”, [(Sathe, Federenko and Zaslavsky, 2024)](https://escholarship.org/uc/item/0qd3662b) as well as [(Elmoznino et al., 2024)](https://arxiv.org/abs/2410.14817), explored in [(Sicilia-Garcia et al, 2002)](https://aclanthology.org/C02-1117/) and in [(Christiansen et al., 2015)](https://www.frontiersin.org/journals/psychology/articles/10.3389/fpsyg.2015.01182/full). This is an emerging formulation of compositionality at the system-level, and refers to the extent to which a language realizes its combinatorial possibilities, which is what we control with $k$. Similarly, we’ll cite the sources and format it in a “Definition” environment. If the reviewer prefers, we can replace formal _compositionality_ with _combinatorial complexity_ or _formal complexity_. That our results show different behavior between semantic and formal compositionality implies that the latter is worth exploration.

---

> ### Author Response · Authors · 2024-11-15
>
> __The first dataset is a very limited Controlled Grammar – much weaker than even a probabilistic context free grammar.__
>
>
> Thank you for this comment. We kindly point out that our grammar is a special case of a PCFG, that generates sentences of fixed length so that we don’t have to control for it in our experiments. The synthetic dataset is strictly necessary for fine-grained control over the degree of compositionality while keeping all else equal: all sentences are constrained to be semantically coherent, to be the same length, preserve the same unigram distribution, and to be extremely unlikely to have been seen during training. These are key _features_ of our setup that required careful design. Moreover, important confounds such as "seen during training" would be impossible to disentangle if we were to construct stimuli from data in-the-wild. Furthermore, we do test inputs the other end of this spectrum with naturalistic in-distribution data, The Pile (see upcoming responses for more details), replicating our key findings with high generality.
>
> We are currently replicating experiments on 3 new grammars, producing different (fixed-)length sentences (will ping once finished). We've already replicated our main results for Llama-3-8B and Mistral-7B, showing these models, like Pythia, differentially code formal complexity and meaning compositionality, see Figs G.2., I.2, and Table 1 in the updated manuscript.

---

> > ### Author Response · Authors · 2024-11-15
> >
> > __I would have expected that a study of compositionality would relate the representation of parts (e.g. words or phrases) to larger constructions (e.g. sentences or paragraphs).__
> >
> > This relation is exactly what we ablate when we shuffle the sequences.

---

> > > ### Author Response · Authors · 2024-11-15
> > > **Sampling distribution**
> > >
> > > __authors say that k contiguous words are coupled during sampling, but they do not describe the sampling distribution… What is the sampling distribution and process for generating data? Is an n-gram language model used? Or a neural model with a given window?__
> > >
> > > We apologize this wasn’t more clear. In brief, our grammar is a PCFG with probability 1 over production rules and uniform probabilities over the leaves. Words are sampled from uniform distributions over a pre-defined vocabulary. Formally,
> > >
> > >
> > > Let there be $l$ variable words in the sequence (these are the colored words in the grammar in Section 3.1). Each position $i = 1\cdots l$ is associated with a vocabulary $\mathcal V^{(i)}$ which contains vocabulary items $v_j^{(i)}$, $j=1\cdots |\mathcal V^{(i)}|$. We set $|\mathcal V^{(i)}|=50$ (line 153) for all $i$. All words in $\mathcal V^{(i)}$ are listed in Appendix E.
> > >
> > >
> > > We state the sampling procedure, illustrated in Fig 1., as follows.
> > >
> > >
> > > $k=1$: for each position $i$ in the sentence, sample the word $w_{i} \sim \text{Unif}( \mathcal V^{(i)})$.
> > >
> > >
> > > $k=2$: couples the vocabularies over bigrams in the sentence. Let $\circ$ be a concatenation operator. We construct a new _coupled_ vocabulary $\mathcal V^{(i,i+1)}$ consisting of bigrams $v_j^{(i,i+1)} := v_j^{(i)}\circ v_j^{(i+1)}$, $j=1\cdots |\mathcal V^{(i)}|$. Then, for $i=1, 3, 5, \cdots l-1$, sample bigrams $w_{i}\circ w_{i+1} \sim \text{Unif}({\mathcal V^{(i,i+1)}})$.
> > >
> > >
> > > $k=3$: couples trigrams in the sentence. Similarly, a new vocabulary over trigrams is constructed: $\mathcal V^{(i,i+1,i+2)}$ consists of trigrams $v_j^{(i,i+1,i+2)} := v_j^{(i)}\circ v_j^{(i+1)}\circ v_j^{(i+2)}$. For $i=1, 4, 7, \cdots l-2$, sample trigrams $w_{i}w_{i+1}w_{i+2} \sim \text{Unif}({\mathcal V^{(i,i+1,i+2)}})$.
> > >
> > >
> > > And so on. We will state this clearly in Section 3.2.1 in the most reader-friendly way possible.

---

> > > > ### Author Response · Authors · 2024-11-15
> > > >
> > > > __I’m not sure how this is supposed to measure compositionality. I could imagine an experiment to evaluate whether changes in nationality or in job led to systematic and predictable differences in the constructed representation, but that was not present.__
> > > >
> > > >
> > > > Thanks for your comment. Could you please elaborate on such an experiment? Crucially, we designed the stimulus sets, varying coupling length $k$, so that the unigram frequencies of each word in the dataset are preserved. This means that there are no directional changes to e.g., nationality/job word occurrences in each dataset. The only change between $k$-settings is the degree of combinatorial complexity in the dataset, which is precisely what we want to control for.

---

> > > > > ### Author Response · Authors · 2024-11-15
> > > > >
> > > > > __sampling 16 contiguous tokens is strange. The 16 contiguous tokens might span sentences, paragraphs, etc. Hence even the real data felt unclean and not representative…Why not sample utterances from The Pile that are whole sentences instead?__
> > > > >
> > > > > Our goal with The Pile was to (1) proxy in-distribution data as well as possible; such that (2) the sequence length is the same as all other tested stimuli (16 words in our case).
> > > > >
> > > > >
> > > > > Why not whole sentences? Causal LMs are trained on strings that do not necessarily coincide with the end of a sentence. For that reason, when making a statement on __in-distribution data__, it is more naturalistic to sample sequences uniformly from the corpus rather than force them to coincide exactly with the starts and ends of sentences. This reflects our goal to adhere as closely as possible to what the model learned in-distribution. We agree that, perhaps for a different question, it may be more appropriate to consider whole sentences.
> > > > >
> > > > >
> > > > > We’ll include this justification in the manuscript.

---

> > > > > > ### Author Response · Authors · 2024-11-15
> > > > > > **Shuffled distributions have lower ID**
> > > > > >
> > > > > > __Figure 2 – it’s odd to me that shuffled distributions have lower intrinsic dimensionality, even in the unigram case. I would have expected the dimensionality would be higher in the case of shuffling, as the data has fewer constraints.__
> > > > > >
> > > > > >
> > > > > > Thanks for pointing it out! We hope our explanation in the section spanning line 461-485 can already answer this point. In short, your hypothesis that shuffled distribution which allows fewer constraints and thus higher combinatorial complexity causing higher dimensionality is reflected in linear effective dimension $d$ as shown in Figure 2.
> > > > > >
> > > > > >
> > > > > > Following Recatanesi (2021), we argue that intrinsic dimensionality reflects latent semantic structure and $I_d$ is collapsed when there is no semantic meaning, as in shuffled case. We want to highlight our investigation on the timepoint when $I_d$ of shuffled data collapse in line 466-475. The collapse of $I_d$ of shuffled dataset corresponds to the timepoint when the models’ linguistic capabilities sharply rise. This signifies that $I_d$ collapse of shuffled data is a sign that the model is capable of extracting meaningful semantic features.

---

> > > > ### Comment · Reviewer_gP62 · 2024-11-15
> > > >
> > > > Thanks for the clarification. Although it is possible to cast this sampling strategy as a degenerate PCFG, I'd argue that this formulation is misleading. Rather, this is a "block unigram" model: once the block size is selected, each block is assigned completely independently, with no respect to the tree structure around it. Furthermore, this distribution is also strictly weaker than the n-gram Markov language model distributions used successfully years ago, in that it does not capture any correlations across blocks. In short, this distribution has little to do with natural language -- any conclusions drawn on this dataset do not necessary apply to actual natural language.

---

> > > > > ### Author Response · Authors · 2024-11-16
> > > > >
> > > > > Thanks for your quick turnaround!
> > > > >
> > > > > We agree with you that our synthetic dataset departs from in-the-wild linguistic phenomena; we'll flesh out these limitations in the section where we introduce The Pile, as well as the Discussion. However, our intention is to study how specific input properties, like combinatorial complexity, impact LMs’ processing. The controlled grammar isolates these properties in a surgical way, which is key to our contribution.
> > > > >
> > > > > >Although it is possible to cast this sampling strategy as a degenerate PCFG, I'd argue that this formulation is misleading. Rather, this is a "block unigram" model
> > > > >
> > > > > We agree that our grammar is a sort of “block unigram” model, and that it is a simple case of a PCFG– our intention was not to mislead. We don’t call the grammar a PCFG in the manuscript.
> > > > >
> > > > > > this distribution has little to do with natural language
> > > > >
> > > > > We understand your hesitance about data that’s synthetically generated, hence unrepresentative of the natural language distribution. First, we kindly point out that all (unshuffled) stimuli are grammatical, and therefore do constitute natural language.
> > > > >
> > > > >
> > > > > We’re afraid that requiring controlled data to resemble the distribution of natural language may be unrealistic; as
> > > > >
> > > > > - Even a more complex grammar would not be faithful to the natural language distribution.
> > > > > - The more complex the grammar, the harder it is to exercise fine-grained experimental control.
> > > > >
> > > > > While our grammar trades off naturalism for experimental control, note again that our __key findings hold for in-distribution natural language data (The Pile)__.
> > > > >
> > > > > We kindly point out that experimental data need not be ecological to be useful. There is a rich psycholinguistic tradition using synthetic data to reveal how LMs and humans process linguistic inputs (see below for a non-exhaustive list). Our work enters into this paradigm, whereby a minimal set of stimuli are sufficient to reveal some processing phenomena, in our case, differential coding of combinatorial and semantic complexity. We emphasize again that key results hold on both controlled and naturalistic data.
> > > > >
> > > > >
> > > > > Humans:\
> > > > > Kutas and Hillyard, (1980) \
> > > > > Lake,.et al. (2019) \
> > > > > Use of “Jabberwocky” sentences in e.g., Hahne and Jescheniak (2001), Federenko et al., (2010, 2016), Desbordes et al., 2022, inter alia.
> > > > >
> > > > > Language models: \
> > > > > SCAN (Lake and Baroni, 2017) and gSCAN (Ruis et al., 2020) \
> > > > > PCFG SET (Hupkes et al., 2020) \
> > > > > Gulordova et al., (2018)

---

> ### Author Response · Authors · 2024-11-15
>
> __Why use the last token as the representation? Did the authors evaluate how well that reflects the earlier tokens in the string? Could, for instance, the original string be reconstructed with reasonable likelihood given this representation?__
>
> Thanks for this question-- it is an interesting point that not only applies to our work, but to that of an entire line of research. We’ll extend our justification in line 244 with the following reasoning.
>
> Given variable sequence lengths due to the tokenization scheme, it is custom in the literature to take the last token representation as a sequence embedding. The literature consensus is that the last token representation in GPT-like models (similar to the [CLS] token in BERT) acts as a sentence embedding, as it is the only to attend to the whole context, and is the one eventually used for next-token prediction– see (Geva et al., 2022; Dar et al., 2022; Elhage et al., 2021). The last token, residual stream representation also has a privileged interpretation as an iterative refinement of vocabulary distributions (Geva et al, 2022; Belrose et al, 2023; Dar et al, 2022) until the next-token prediction; or a communication channel onto which layers write (Elhage et al., 2021; Merullo et al., 2024).
>
>
> Importantly, it's also the default in the existing literature on ID estimation in LMs. The vast majority of existing works (Cheng et al. 2023; Cai et al. 2021; Chen et al. 2020; Antonello et al., 2024; Doimo et al., 2024; Tulchinskii et al, 2023; Yin et al, 2024) compute geometric measures such as ID at the level of the residual stream, on the last token representation.
>
>
> __On representational faithfulness of the last token embedding:__ This is a great question. We, along with the many works that consider the last-token representation, did not attempt an explicit reconstruction of the original string from the last-token embedding. This would be highly nontrivial as the loss landscape would be nonconvex; any neural approach is likely to find local minima. An entirely standalone line of research is dedicated to inverting LMs, see, e.g. [Zhang et al., 2024](https://arxiv.org/abs/2405.15012), [Morris et al., 2023](https://arxiv.org/abs/2311.13647).
>
>
> But, an explicit reconstruction of the inputs is actually unnecessary, as it’s reasonable to assume an _injective_ mapping from our inputs to last-token embeddings. That is, given two different inputs, it is highly unlikely that their last-token embeddings are exactly the same, even under quantization; this is due to the small size of our input set compared to the (much larger) representational capacity of a hidden layer. Injectivity is attested empirically in our experiments. The TwoNN estimator fails when there are duplicates in the point cloud (the $\mu_i$ would be $\infty$ under the presence of duplicates), and this happened for 0 cases.
>
> Finally, if we want to look at a representation for which there is ``no information leakage", we can consider the last-layer, last-token representation, whose invertibility is highly likely according to the above sources. When only considering this representation, all of our results hold.
>
> We will add this justification in an Appendix section in the paper. Let us know if you have more questions!

---

> > ### Author Response · Authors · 2024-11-15
> >
> > __Figure 2 – is this mostly telling us something about the dimension from which the data was sampled? What happens to these numbers if the number of categories is changed?__
> >
> > Yes exactly! Changing $k$ changes the effective number of categories, or _degrees of freedom_ in the dataset, which we touch on in line 202. We apologize this wasn’t more clear. In brief, recall that for $k=1$, for each position $i$ in the sentence, the word $w_i \sim \text{Unif}(\mathcal V^{(i)})$ is sampled independently. Then, $k=2$ couples bigrams in the sentence, which effectively _halves_ the number of independent categories. Similarly, $k=3$ couples trigrams in the sentence, dividing the number of independent categories by 3…and so on.
> >
> >
> > By increasing $k$, we decrease the number of independent categories, which, in turn, decreases the combinatorial complexity (what we call formal compositionality) of the dataset. Our key result is that LMs _preserve_ this complexity ordering in the dimension of their representations.
> >
> > We’ll add this explanation in the paper.
> >
> >
> > More along these lines, we are introducing new experiments for grammars of different lengths (another way to change the number of categories). Stay tuned for those results.

---

> ### Author Response · Authors · 2024-11-23
>
> Thank you for your comments and feedback again.
> As the end of the discussion period is approaching, we would like to hear your further thoughts on our response and additional results summarized on the top as a global response.

---

> ### Author Response · Authors · 2024-12-02
>
> Thank you for your feedback again, this is our last reminder again for your further thoughts and comments.
> We hope our response addressed your questions and concerns and we would greatly appreciate it if you were willing to consider increasing your score if you are satisfied with our response.

---

### Official Review · Reviewer_DUDX · 2024-11-08

**Soundness:** 2
**Presentation:** 4
**Contribution:** 3
**Rating:** 8
**Confidence:** 3

**Summary:**

This paper analyses the representations of the last token in sentences from a pidgin language constructed by the authors.

Specifically the authors vary three things on the network side and two things in the input:

1. network training progress: Networks trained for longer are supposed to be better/sophisticated
2. network layer: Later network layers are supposed to be more important for "semantics" compared to "form"
3. Network size: Representations from bigger networks are supposed to be "better"
4. sentence word order: This is quantified as a binary variable, whether sentences are shuffled therefore destroying "semantics" or not
5. inter-word coupling: The variation in input sentence's complexity due to coupling adjacent words is quantified by using gzip compressed size to estimate the Kolmogrov complexity.

They randomly sample sentences from their pidgin language and they measure two statistics for the last token's embedding vector
1. Two nearest neighbor estimator for manifold dimension: An estimate of the dimensionality of the non-linear manifold under some assumptions.
2. PCA dimensionality for 99% variance: The dimensionality of the linear subspace

So we can now imagine a 7-column dataframe with 5-dimensions and 2-measures and the paper presents various interesting observations.

The main claim from the authors are that: 1) "PCA dimension" of final token representation is a good estimate of kolmogrov/combinatorial complexity, 2) "TwoNN dimension" is a good estimate of "semantic" complexity. 3) The TwoNN complexity vs layer curve is a good predictor of how well the network is trained. If TwoNN complexity is higher for later layers then the model is trained well otherwise not.

**Strengths:**

The experiments in this paper are quite interesting, novel and certainly thought provoking. This paper presents correlates statistics of LLM representations to input complexity, and it disentangles input complexity into "form" and "meaning" which is a very neat idea. The paper is well-written and clearly organized. The introduction provides a strong motivation for the research and effectively sets the stage for the key questions addressed.

**Weaknesses:**

This is a tough paper to assess because the connection between the conclusions and claims in the paper , and the actual experimental observations is a bit speculative.

For example, one of the claims is that shuffling words destorys any semantic information and therefore the TwoNN dimensionality is lower when we shuffle words, compared to when we do not. But this phenomenon is observed for 3 out of 4 settings, and the red curves in figure 2 where words coupled are 4 do not show that behavior.

The main issue is that this is a highly empirical paper and there is a danger that the results come from "data fishing" by slicing and dicing a fairly complex dataframe. The overall story is plausible and definitely thought provoking but I am not sure how conclusive the evidence in the paper is.

Despite these weaknesses, I think the ideas in the paper are certainly very interesting, even though

**Questions:**

The phase transition in figure three seems to explain performance for only a few tasks such as SciQ, ArcEasy, Lambada, and PIQA but not for other tasks. Why do you think that is?

---

> ### Author Response · Authors · 2024-11-15
>
> Thanks so much for your careful review. We’re glad you found the insights in our paper to be interesting, and thanks for your constructive suggestions for improvement. We’re responding to each comment below (with more experiments to come, stay tuned).

---

> ### Author Response · Authors · 2024-11-15
>
> __...one of the claims is that shuffling words destorys any semantic information and therefore the TwoNN dimensionality is lower when we shuffle words, compared to when we do not. But this phenomenon is observed for 3 out of 4 settings, and the red curves in figure 2 where words coupled are 4 do not show that behavior.__
>
> Thanks for this comment; we agree and we will reformulate our claim to be weaker as follows:
>
> Shuffling, which destroys sentence-level semantics, collapses the ID to a narrow range. Now, the only differentiator between the ID curves is formal, and not due to sentence-level semantics. For datasets of higher semantic complexity in the normal/sane case (k=1..3), the ID is lower upon shuffling.
>
> We’ll change the text throughout the manuscript to reflect this.

---

> ### Author Response · Authors · 2024-11-15
>
> __...a danger that the results come from "data fishing" by slicing and dicing a fairly complex dataframe.__
>
> Thanks for raising this concern. It is unfortunately highly nontrivial to cover all possible syntactic structures, but to alleviate this, we’re expanding the experiments by introducing new grammars that generate sentences of different lengths, still with tunable complexity via $k$. Stay tuned for those results.
>
> To strengthen our result, so far we’ve reproduced the current experiments on two different LMs, Llama-3-8B and Mistral-7B.
> We replicated the tendency for (1) $I_d$ and $d$ to decrease as $k$ increases; (2) $I_d$ to collapse to a narrow range upon shuffling, while $d$ increases to a narrow range; (3) formal complexity (gzip) to highly correlate to $d$, but not to $I_d$.
>
> We will soon post a new version of the pdf where the analogue to Fig 4 for Mistral and Llama is found in Fig G.2, and the new Kolmogorov complexity correlations per layer in Fig. I.2.

---

> > ### Author Response · Authors · 2024-11-15
> > **Update: See Fig G.2; Fig. I.2; Table 1 for Llama and Mistral replication study**
> >
> > We just uploaded a new version of the manuscript with results on Llama-3-8B and Mistral-7B. We did the final training checkpoint experiments only, replicating Fig 4, now in Fig G.2., and Fig I.1., now in Fig I.2. We added the correlation between $d$ and gzip to Table 1.

---

> ### Author Response · Authors · 2024-11-15
>
> ___The phase transition in figure three seems to explain performance for only a few tasks such as SciQ, ArcEasy, Lambada, and PIQA but not for other tasks. Why do you think that is?___
>
> Great question! This is likely because the other tasks are too hard. If you take a look at Fig 3, in all plots, the tasks that don’t improve stay at baseline performance (e.g., Logiqa, orange). In addition, if you compare the performance of the bigger/better models to the smaller ones (right -> left), there are tasks like ArcChallenge (red) and WinoGrande (blue) for which performance remains at baseline for the smallest model, and increases for the biggest model starting at the ID phase transition.
>
> We’ll include this discussion in an Appendix.

---

> ### Author Response · Authors · 2024-11-23
>
> Thank you for your comment and feedback again!
> As the discussion period is approaching to its end, we are writing a reminder that we added additional experiments that strenghthen our claim and looking forward to your further thoughts.

---

### Author Response · Authors · 2024-11-20
**New experiments**

We thank all the reviewers for their insightful comments and suggestions. In response to your reviews, we have __increased the number of experiments by ~6x__, which are detailed below and can be found in the updated PDF.

## Llama and Mistral

First, following R3’s suggestion, we ran the full suite of experiments for Llama-3-8B and Mistral-7B; all results hold (see Fig. I.3).
\


## New grammars
We thank R2 and R3 for suggesting to diversify the synthetic datasets. We have replicated all experiments on all fully trained models, including Llama and Mistral, on _4 more grammars of length 5, 8, 11, and 15_. For maximal comparability, new grammars have a similar structure to our original one, but with fewer semantic categories (see details in the updated Appendix E). Existing results generalize to the new datasets.

Results can be summarized as follows.

1. __Feature complexity increases with sequence length (see Figure J.1):__ both nonlinear and linear feature complexity increase with sequence length. Moreover, all curves saturate, or plateau, around length=11, indicating this dependence is sublinear.

2. __Differential coding of semantic complexity emerges with increasing sequence length (see Figure J.3)__: for the shortest sequence length of 5, the difference between the $I_d$ of the coherent and shuffled text is around 0, suggesting that at shorter lengths, the semantic complexity of a grammatical sentence proxies that of a bag of words, probably because the semantics of shorter sentences are very simple. This suggests the difference in semantic complexity between the coherent and shuffled text is an emergent property of sentence length, as the semantics of longer sentences become more complex.


3. __Kolmogorov complexity experiments are robust to sequence length (see Table I.1 and Figure I.4):__ our initial conclusions from the Kolmogorov complexity/gzip experiments which were originally run on the length 17 grammar remain true for grammars of all lengths.

Overall, we thank the reviewers for suggesting these additional experiments. We believe they confirm the robustness of our hypotheses to different datasets, while also revealing an interesting additional relationship between feature complexity and sequence length, which strengthens our findings.

Finally, thanks to R2, R4, and R5 for suggesting to better contextualize our compositionality definitions in the literature. We will add the relevant discussion and citations (see your individual responses) to the Related Work and Methods sections. We hope you agree that this discussion is sufficient to motivate our separation of formal and meaning complexity.

---

### Note · Authors · 2024-12-14

**Comment:**

We thank all the reviewers again. We reached to the conclusion that reframing the meaning of compositionality would greatly benefit our argument.
We plan to improve our paper and resubmit to another venue, and thus withdraw from ICLR 2025.

**Withdrawal Confirmation:**

I have read and agree with the venue's withdrawal policy on behalf of myself and my co-authors.